# Study on Fatigue Cracking of Diaphragm’s Arc Opening of OSD in Steel Bridges by Using Biaxial Stress Method

**DOI:** 10.3390/ma16155217

**Published:** 2023-07-25

**Authors:** Yong Zeng, Hongtao Kang, Xueqin Li, Zhijie Li, Yunchuan Xiao, Jianting Zhou

**Affiliations:** 1State Key Laboratory of Mountain Bridge and Tunnel Engineering, Chongqing Jiaotong University, Chongqing 400074, China; m13243566929@163.com (H.K.); m13451163369@163.com (X.L.); 15723097894@163.com (Z.L.); m1760833081@163.com (Y.X.); jt-zhou@163.com (J.Z.); 2Mountain Bridge and Materials Engineering Research Center of Ministry of Education, Chongqing Jiaotong University, Chongqing 400074, China

**Keywords:** orthotropic steel deck (OSD), diaphragm plate, stress composition, moving load, U-rib, arc opening

## Abstract

Changes in loading position have a significant impact on the stress field of each vulnerable area of an orthotropic steel deck (OSD). The arc opening area of the diaphragm and the connecting area between the U-rib and the diaphragm under the moving load are prone to fatigue cracking. By comparing the stress responses under different methods, the hot spot stress (HSS) method is used as the main stress extraction method in fatigue performance evaluation. The control stress of fatigue cracking was analyzed by comparing the direction of the principal stress field with the crack direction in this experiment. According to the stress amplitude deviation under the biaxial stress state, a set of methods for evaluating the effects of in-plane biaxial fatigue was developed. An improved luffing fatigue assessment S–N curve was applied to analyze the fatigue life of the diaphragm’s arc opening area. The results show that when the moving load is exactly above the connection of the deck and the web of the U-rib on one side, it is in the most unfavorable position in the transverse direction, and the diaphragm is mainly under the in-plane stress state. The longitudinal range of the stress influence line of the arc opening is approximately twice the diaphragm spacing. Two to three stress cycles are caused by one fatigue load. Fatigue crack control stress is the principal stress tangential to the arc opening’s edge in this area. The normal direction of the principal stress in the model test is roughly consistent with the crack initiation direction. The variation in the stress amplitude deviation in this area is caused by changes in the action position of the moving load. When the moving load is at a certain distance from the involved diaphragm, it is reduced to zero, implying that the in-plane fatigue effect is the greatest in this area.

## 1. Introduction

The orthotropic steel deck (OSD) is frequently used in long-span steel bridges owing to its light weight, high strength characteristics, and excellent stress properties. Figure 1 shows the main components of the OSD. The diaphragm’s arc opening area is one of the key fatigue-prone details of an OSD [1]. Fatigue cracking in the diaphragm’s arc opening area often occurs at the edge of the opening. The stress distribution of the curved cut-outs is closely related to geometrical parameters such as the diaphragm’s arc opening pattern, the web thickness of the diaphragm and longitudinal ribs, the presence or absence of an inner diaphragm, and the welding quality [2,3].

The diaphragm–U-rib joint is a complex and fatigue-prone part, and the stress concentration at the weld toe is significant under a moving load [4,5,6,7]. In addition, ensuring welding quality in actual structures is difficult, which may result in fatigue cracking due to possible welding defects. According to recent research findings, the relative structural stiffness of the diaphragm web and U-rib, the diaphragm hole-digging type, and the welding process of their connecting parts all have a significant impact on fatigue performance [8,9].

Cuninghame J.R. [10] first made a deep mechanism analysis of the causes of fatigue cracking in the vulnerable area of the structure of Severn Bridge in the UK in 1987, and summarized the vulnerable parts of the bridge deck with a fatigue cracking risk based on the calculation results. Robert J. Connor [11] discussed the stress response of orthotropic plates under load in different test environments, carried out long-term stress monitoring on the Williamsburg bridge, and summarized the stress amplitude of each vulnerable area from the data in 2002. Michèle S. Pfeil [12] obtained the stress concentration factor in the local area of a trapezoidal stiffener based on the measured data of test strain gauges and FEM (finite element method) simulation results in 2005. John R. Fisher [13] obtained the stress amplitude of each fatigue detail area of an actual bridge based on field-measured fatigue test data and then evaluated its fatigue life in 2005. Zhi-Gang Xiao [14] discussed the fatigue performance of a U-rib butt joint area under constant stress amplitude and compared the test results with the values calculated based on the linear elastic fracture mechanics theory. Mustafa Aygül [15] predicted the fatigue life of welded joints between the longitudinal ribs and diaphragms of orthotropic steel bridge decks in 2012 using common steel bridge members’ fatigue strength assessment methods. Liu [16] studied the fatigue performance of the connection area between a deck roof and a U-rib, extracted the stress data using the method of linear extrapolation of HSS, and focused on the impact of the difference in moving load location in the transverse direction in 2015. PIETRO [17] evaluated the fatigue strength of the longitudinal rib diaphragm connection joint, deduced the mathematical formula used to express the stress intensity factor K by using the Paris law and the J-integral method, and obtained the actual fatigue life by extrapolating the measured data and theoretical calculation values. Karlo [18] studied the prediction of residual stress and structural deformation caused by the welding process during manufacturing. The data were compared based on the butt welding of two plates and the T-joint fillet welding of two plates.

However, the cracking mechanism of an OSD with an arc opening has not been explored yet, and such fatigue cracks are observed during the relevant fatigue tests. Therefore, the fatigue performance of a diaphragm’s arc opening area based on this phenomenon of fatigue cracking is analyzed in this paper. In this paper, the stress response of the diaphragm’s arc opening area under the local action of the moving loads is obtained by modeling and analysis using FEM software known as ABAQUS (2018). The stress composition of the diaphragm’s arc opening area and the control stresses under the corresponding fatigue failure mode can be analyzed based on the stress responses. The fatigue properties and reasons for cracking in this area are also discussed. The fatigue effects of the diaphragm and its influencing factors are analyzed using the fatigue effect evaluation criteria. The research details on the fatigue life are analyzed based on the improved variable amplitude fatigue assessment S–N curve.

## 2. Fatigue Assessment Methods of OSD

### 2.1. HSS Method

Hot spot stress (HSS) refers to the maximum structural stress or the stresses at dangerous points on a dangerous section in a structure [19,20]. It includes membrane stress and bending stress, but nonlinear stress is not considered. The HSS value for fatigue assessment can be accurately and rapidly extracted only by reasonably dividing the mesh based on the specific structural form and solution requirements.

The recommended extrapolation rules for HSS are mentioned in the evaluation codes of various countries. The extrapolation method is illustrated in Figure 2. Some of the commonly used specifications are as follows [21,22,23,24]:

#### 2.1.1. International Welding Society (IIW) Recommended Method

For two-point linear extrapolation calculation, the stress extraction points at 0.4 *t* and 1.0 *t* from the weld toe are as follows:(1)$$\sigma_{hss}=1.67\sigma_{0.4t}-0.67\sigma_{1.0t}$$

σhss—hot spot stress;
*t*—plate thickness.

For three-point linear extrapolation calculation, the stress extraction points at 0.4 *t*, 0.9 *t*, and 1.4 *t* from the weld toe are as follows:(2)$$\sigma_{hss}=2.52\sigma_{0.4t}-2.24\sigma_{0.9t}+0.72\sigma_{1.4t}$$

$\sigma_{hss}$—hot spot stress;
*t*—plate thickness.

#### 2.1.2. Det Norske Veritas Recommended Method

For two-point linear extrapolation calculation, the stress extraction points at 0.5 *t* and 1.5 *t* from the weld toe are as follows:(3)$$\sigma_{hss}=1.5\sigma_{0.5t}-0.5\sigma_{1.5t}$$

σhss—hot spot stress;
*t*—plate thickness.

For three-point linear extrapolation calculation, the stress extraction points at 0.5 *t*, 1.5 *t*, and 2.5 *t* from the weld toe are as follows:(4)$$\sigma_{hss}=3.75\sigma_{0.5t}-2.5\sigma_{1.5t}+0.375\sigma_{2.5t}$$

σhss—hot spot stress;
*t*—plate thickness.

The reference point is selected to be the stress extraction point 0.5 *t* from the weld toe:(5)$$\sigma_{hss}=1.125\sigma_{0.5t}$$

$\sigma_{hss}$—hot spot stress;
*t*—plate thickness.

#### 2.1.3. HSS Analysis Process

The hot spot types can be roughly categorized into the following three categories: A, B, and C, as shown in Figure 3, for the stress analysis of the weld toe area of welded structures.

On the motherboard, near the junction where the attached board’s end meets the motherboard, are type A hot spots. Type B [25] hot spots can be noted on the attachment board, in the same area as the type A hot spots. Type C hot spots can be found on either the motherboard or the attachment board in the non-end motherboard junction area of the attachment plate.

The type A and C hot spots can be determined using the above Equations (1)–(5). The thickness of the steel component is not discussed in the stress analysis because the type B hot spot’s crack growth mode gradually extends along the edge of the steel component. The following is a quick explanation of the calculation concept [26].

When the mesh division is fine, the distances of the stress extraction points are taken at 4 mm, 8 mm, and 12 mm. The HSS value is then calculated using the following formula using three reference points (6):


(6)
$$\sigma_{hss}=3\sigma_{4\mathrm{mm}}-3\sigma_{8\mathrm{mm}}+\sigma_{12\mathrm{mm}}$$


$\sigma_{hss}$—hot spot stress;$\sigma_{4\mathrm{mm}}$—stress at 4 mm;$\sigma_{8\mathrm{mm}}$—stress at 8 mm;$\sigma_{12\mathrm{mm}}$—stress at 12 mm.

2.When the mesh division is rough, the distances of the stress extraction points are taken at 5 mm and 15 mm. Then, the HSS value is calculated using the following formula using two reference points (7):


(7)
$$\sigma_{hss}=1.5\sigma_{5\mathrm{mm}}-0.5\sigma_{15\mathrm{mm}}$$


$\sigma_{hss}$—hot spot stress;$\sigma_{5\mathrm{mm}}$—stress at 5 mm;$\sigma_{15\mathrm{mm}}$—stress at 15 mm.

### 2.2. Fatigue Stress Evaluation Criteria

#### 2.2.1. Stress Amplitude Deviation Value

The principal stress direction and the crack propagation direction are typically parallel [27]. When the principal stress is the major stress component, the crack propagation direction is perpendicular to the coordinate axis of the established reference coordinate system. The structure is currently under a complex multiaxial stress state, which deviates from this ideal scenario. The relationship between fatigue crack growth, the primary stress, and the major stress components needs to be analyzed. The numerical changes in the primary stress and the major stress components are used to assess the spatial fatigue effect of the local area. Therefore, in the numerical analysis, the method of the absolute maximum principal stress is used.

The calculation and analysis of fatigue cracking in steel plates fall within the elasticity range, so the following formula can be used to solve for the principal stress at a point in three dimensions using the elasticity theory [27]. The symbols in the formulas in this section have the same meanings as those in linear elasticity.
(8)$$\begin{array}{c} \sigma^{3}-\varphi_{1}\sigma^{2}+\varphi_{2}\sigma -\varphi_{3}=0\\ \varphi_{1}=\sigma_{x}+\sigma_{y}+\sigma_{z}\\ \varphi_{2}=\sigma_{x}\sigma_{y}+\sigma_{y}\sigma_{z}+\sigma_{x}\sigma_{z}-\tau_{xy}{}^{2}-\tau_{yz}^{2}-\tau_{zx}{}^{2}\\ \varphi_{3}=\sigma_{x}\sigma_{y}\sigma_{z}-\sigma_{x}\tau_{yz}{}^{2}-\sigma_{y}\tau_{zx}{}^{2}-\sigma_{z}\tau_{xy}{}^{2}+2\tau_{xy}\tau_{yz}\tau_{zx} \end{array}$$

σ is the principal stress;$\varphi_{1}\sim \varphi_{3}$ is the stress tensor invariant.

Both the principal stress and the major stress components are vectors. The relationship between the principal stress and the major stress components includes the magnitude and direction of the stresses. The formula for solving the cosines of the principal stress directions at a certain point in elasticity is as follows [27].
(9)$$\left\{\begin{array}{c} \left(\sigma_{x}-\sigma_{i}\right)l_{i}+\tau_{yx}m_{i}+\tau_{zx}n_{i}=0\\ \tau_{xy}l_{i}+\left(\sigma_{y}-\sigma_{i}\right)m_{i}+\tau_{zy}n_{i}=0\\ \tau_{xz}l_{i}+\tau_{yz}m_{i}+\left(\sigma_{z}-\sigma_{i}\right)n_{i}=0\\ l_{i}^{2}+m_{i}^{2}+n_{i}^{2}=1 \end{array}\right.$$

$l_{i}$, $m_{i}$ and $n_{i}$ are the direction cosines of principal stresses $\sigma_{i}$, where *i* = 1~3.

According to Equation (9), if $l_{i}$, $m_{i}$, and $n_{i}$ are the solutions of the equations, then $-l_{i}$, $-m_{i}$, and $-n_{i}$ can also be the solutions of the equations. The actual calculation can be carried out by using positive numbers. Equation (10) illustrates the mathematical expression of a positive numerical solution.
(10)$$\left\{\begin{array}{l} l_{i}=\frac{A_{i}}{\sqrt{A_{i}^{2}+B_{i}^{2}+C_{i}^{2}}}\\ \;m_{i}=\frac{B_{i}}{\sqrt{A_{i}^{2}+B_{i}^{2}+C_{i}^{2}}}\\ n_{i}=\frac{C_{i}}{\sqrt{A_{i}^{2}+B_{i}^{2}+C_{i}^{2}}} \end{array}\right.$$

In Equation (10):(11)$$\left\{\begin{array}{c} \tau_{xy}=\tau_{yz}=\tau_{zx}=0\\ B_{i}=\tau_{xy}\tau_{zx}-\left(\sigma_{x}-\sigma_{i}\right)\tau_{yz}\\ C_{i}=\left(\sigma_{x}-\sigma_{i}\right)\left(\sigma_{y}-\sigma_{i}\right)-\tau_{xy}^{2} \end{array}\right.$$

The formula above is straightforward, but it is not a universal one. It is not applicable if all of $\tau_{xy},\;\tau_{yz},\;\tau_{zx}$ are zero or if two of them are zero. If either $\tau_{xy}=\tau_{yz}=\tau_{zx}=0$ or both of them are zero, the following is the special solution.

If $\tau_{xy}=\tau_{yz}=\tau_{zx}=0$, then $\sigma_{x},\;\sigma_{y},\;\sigma_{z}$ are the principal stresses. In this case, Equation (9) changes to the following:(12)$$\left\{\begin{array}{c} \left(\sigma_{x}-\sigma_{i}\right)l_{i}=0\\ \left(\sigma_{y}-\sigma_{i}\right)m_{i}=0\\ \left(\sigma_{z}-\sigma_{i}\right)n_{i}=0\\ l_{i}^{2}+m_{i}^{2}+n_{i}^{2}=1 \end{array}\right.$$

If $\sigma_{i}=\sigma_{x}$, then $l_{i}=1,m_{i}=n_{i}=0$ of $\sigma_{i}$.

If σi=σy, then mi=1,li=ni=0 of σi.

If $\sigma_{i}=\sigma_{z}$, then $n_{i}=1,l_{i}=m_{i}=0$ of $\sigma_{i}$.

The above solution formula should be used to avoid a situation in which any two principal stresses are equal in $l_{i},\;m_{i},\;n_{i}$ due to their equal principal stress values when the three principal stresses are equal or when both of them are equal.

3.If $\tau_{xy}=\tau_{yz}=0,\tau_{zx}\neq 0$, then $\sigma_{y}$ is the principal stress. Under this condition, Equation (9) becomes the following.




(13)
$$\left\{\begin{array}{c} \left(\sigma_{x}-\sigma_{i}\right)l_{i}+\tau_{zx}n_{i}=0\\ \left(\sigma_{y}-\sigma_{i}\right)m_{i}=0\\ \tau_{xz}l_{i}+\left(\sigma_{z}-\sigma_{i}\right)n_{i}=0\\ l_{i}^{2}+m_{i}^{2}+n_{i}^{2}=1 \end{array}\right.$$



If $\sigma_{i}=\sigma_{y}$, then $m_{i}=1,l_{i}=n_{i}=0$ of $\sigma_{i}$.

If $\sigma_{i}\neq \sigma_{y}$, then $m_{i}=0$ of $\sigma_{i}$, $k_{i}=-\frac{\left(\sigma_{x}-\sigma_{i}\right)}{\tau_{xz}}$ is achieved through Equation (13) to obtain $l_{i}=\frac{1}{\sqrt{1+k_{i}^{2}}},m_{i}=0,n_{i}=k_{i}l_{i}$ of $\sigma_{i}$.

The cases of $\tau_{zx}=\tau_{yz}=0,\tau_{xy}\neq 0$ and $\tau_{zx}=\tau_{xy}=0,\tau_{yz}\neq 0$ are the same and are not repeated herein.

#### 2.2.2. Uniaxial Load

Steel specimens subjected to uniaxial loads only bear unidirectional normal or shear stress. It is advised to set the normal stress as $\sigma_{x}$ by the selection of the reference coordinate system if it only bears uniaxial normal stress. $\sigma_{x}$ could also be used as the main stress component. The three-dimensional principal stress values are $\sigma_{x}$, 0, and 0, respectively, when $\sigma_{x}\neq 0$ is added to the previous Equation (8). The principal stress with the largest absolute value is $\sigma_{x}$. The direction cosine value corresponding to $\sigma_{x}$ is solved by substituting $\sigma_{i}=\sigma_{x}$ into the previous Equation (9): ±1, 0, 0.

If the specimen is applied only to uniaxial shear stress and is in a pure shear stress state, the shear stress may be set to $\tau_{xy}$. $\sigma_{x1}$ in the direction of reference coordinate axis $x_{1}$ is the main stress component. After coordinate transformation, $\sigma_{x1}=\tau_{xy}$ is obtained.

The three-dimensional principal stress values are $\tau_{xy}$, 0, and $-\tau_{xy}$, respectively, when $\tau_{xy}\neq 0$ is added to the previous Equation (8). The principal stress with the largest absolute value is obviously $\tau_{xy}$. $\sigma_{i}=\tau_{xy}$ is added to the previous Equation (9).

The direction cosine value is then solved, corresponding to $\tau_{xy}$: $\frac{\sqrt{2}}{2},\;\frac{\sqrt{2}}{2},\;0$, $-\frac{\sqrt{2}}{2},\;-\frac{\sqrt{2}}{2},\;0$.

#### 2.2.3. Biaxial Stress State

In the actual operation stage, steel members are typically in a biaxial stress state. The maximum principal stress deviates from the major stress components, and at least two of the six stress components are not zero. The ratio of the maximum value of stress amplitude is used to quantify the deviation between the maximum principal stress and the major stress components because the maximum value of stress amplitude $\Delta_{\sigma }$ accounts for the largest proportion of fatigue damage in one stress process. The introduction of deviation value δ [27] is as follows:(14)$$\delta =\frac{\Delta_{\sigma_{n}}}{\Delta_{\sigma_{m}}}$$

$\Delta_{\sigma_{n}}$ is the maximum major stress amplitude at all levels and $\Delta_{\sigma_{m}}$ is the maximum principal stress amplitude in the formula above. The expression for δ shows that the deviation between $\sigma_{m}$ and $\sigma_{n}$ increases as the difference between δ and 1 grows. If δ=1, it is included in the category of uniaxial fatigue.

The above analysis shows that the major stress component is the principal stress with the largest absolute value under uniaxial stress. Under the biaxial stress state, the principal stress with the largest absolute value and the major stress component is typically different. The larger the gap, the more severe the multiaxial fatigue effect is.

According to the analysis in Section 5, the diaphragm can be regarded as a thin plate. The second and third principal stresses are located within the thin plate and are in-plane stresses. As shown in Figure 4, $\sigma_{principal1}$ points towards the Z direction of the thin plate, which is the out-of-plane stress, basically 0. Under biaxial compression, the diaphragm can be considered as being in a biaxial stress state. The fatigue crack trend at the curved openings of the diaphragm is not perpendicular to the Y-axis. This reveals that using the stress component in the Y direction to evaluate fatigue performance is ineffective. $\sigma_{principal3(\max)}$ should be used for evaluation, approximately perpendicular to the direction of fatigue crack growth. Additionally, the stress amplitude deviation can be used to evaluate the spatial fatigue effect of changing moving load positions.

### 2.3. Finite Element Modeling Method

By using finite element simulation calculation methods, the accuracy of the HSS method and fatigue stress assessment criteria can be improved. The different parameters of the finite element simulation method have a significant impact on the simulation calculation results.

#### 2.3.1. Modeling Process

The U-rib’s size and diaphragm’s arc opening form used in the numerical simulation in this section are consistent with those used in the fatigue test in the previous section. The size parameters of the U-rib are highlighted in Figure 5.

The model has three transverse diaphragms running longitudinally and seven U-ribs running transversely. ABAQUS (2018) FEM software is used for modeling and analysis. Figure 6 shows the FEM model and constraints. The top plates, U-ribs, and diaphragms in the model are made of C3D8R solid elements with a mesh size of 20 mm. The joints between the U-ribs and diaphragms are modeled as common nodes, and the element mesh is densified around the relevant details with a mesh size of 1 mm. The boundary condition is that the longitudinal ends are simply supported. The diaphragm is restrained by six degrees of freedom. The material in the model is Q345D steel. The steel’s density is 7850 kg/m^3^, the steel elastic modulus, E, is 206 GPa, and the Poisson’s ratio, µ, is 0.3.

#### 2.3.2. Selection and Application of Fatigue Loads

The local stress effect of a moving load on the structural details of an OSD in the vertical and horizontal directions is prominent, which can be seen from the existing research results. The fatigue load is based on the principle of loading a single fatigue vehicle. During the FEM analysis, special attention is paid to the local fatigue-vulnerable area of the structure under the most unfavorable longitudinal and transverse loading of the fatigue vehicle.

The fatigue evaluation herein is based on AASHTO LRFD [28]. Figure 7 represents the simplified standard fatigue vehicle HS15 in this specification. The transverse track width of the fatigue car is 1.8 m, the middle and rear axles both weigh 108 kN, and the load area is 0.51 m (transverse) × 0.25 m (longitudinal). Furthermore, the vehicle impact coefficient is considered to be 0.15 for the fatigue load.

Since the calculation model and fatigue test model have the same structural design parameters, the application size and action area of the standard vehicle moving load are scaled according to the corresponding proportion (1:2). The actual stress response results of the structural details are determined. The following analysis depicts that the stress influence line of the vulnerable area of the curved cuts of the diaphragm is relatively short. The distance between the adjacent diaphragms of the model is 1.35 m, which is less than the wheelbase of the front and middle axles of the fatigue vehicle, and is 4.5 m less than the wheelbase of the middle and rear axles (the size of the fatigue vehicle is scaled with the scale of the test model). Therefore, the moving load is only loaded by the middle axle of the fatigue vehicle. The contact area after the moving load diffuses to the bridge deck at 45 degrees is 0.305 m × 0.175 m, and the wheel pressure value is 291 kPa.

## 3. Model Experiment of OSD

### 3.1. Experiment Model

The scale of the experiment model is 1:2, wherein seven longitudinal ribs are included in the transverse direction, three diaphragms are considered in the longitudinal direction, and 10 mm thick steel plate heads are at the end. The spacing of each diaphragm is 1.35 m, the thickness of the deck’s top plate is 7 mm, the thickness of U-rib web is 4 mm, and the thickness of the diaphragm stiffener is 5 mm. A 120.9 mm high, 2400 mm long, and 450 mm wide steel box was fabricated to support the bridge deck structure to keep it in the horizontal plane. To approximate the actual stress condition of the OSD model, a trapezoidal steel plate with welded side ribs was spliced at the bottom of the middle diaphragm HGB2. The structural size selection and boundary condition setting were designed according to the basic principle of similarity theory. The specific size design of the experiment model is represented in Figure 8, and the installation picture of the experiment model is given in Figure 9.

### 3.2. Arrangement of Strain Measuring Points of Experimental Model

The strain gauges at the measuring points of the steel deck model were mainly placed in the webs, the diaphragm plates, the U-ribs, and the deck plates. At the bottom of the model, displacement measuring points were also placed. The experimental model has 157 strain gauges and 6 displacement measuring points installed. The structural details of the middle diaphragm (HGB2) are mainly noticeable since the points with large stresses are typically located in the webs of the middle diaphragm.

The layout of the stress measuring points on the web of diaphragm plates is shown in Figure 10a. The diaphragm’s longitudinal and transverse stresses are close to one another, according to preliminary calculation results, which can be expressed as a planar effect. As a result, three-way strain rosettes at a 45-degree angle were mainly set up. Auxiliary measuring points were added on the trapezoidal plate to help with the analysis in order to eliminate the diaphragm plate’s warping effect at the bottom of HGB2 and to accurately simulate the actual bridge.

Figure 10b shows the layout of the stress measuring points on the U-ribs. The one-way measuring point was adopted at the bottom and the three-dimensional strain rosette was adopted at the top of the U-ribs due to the obvious tension at the bottom and the complex stresses at the top. A unidirectional strain gauge was installed at the wheel compression edge because the deck plate showed local large unidirectional tensile and compressive stresses as it was subjected to moving loads. To measure the deflection changes, three electronic displacement indicators, S01, S02, and S03, were arranged at an equal distance at the bottom.

### 3.3. Experimental Loading Process

The bridge deck of the experimental model is first applied to a secondary-stage dead load of 15 kN, and a live load of 47.7 kN is then simulated and applied. MTS loads the secondary dead load and the live load.

(1)Static Load Test Process

Before loading, it is necessary to run a number of static preloading tests to guarantee a smooth test. After the fatigue load cycle has reached the required number of times, the machine is stopped for a static load test in order to collect the stress. The test’s maximum static load is 91.2 kN. The process of loading and unloading a load is as follows: 0 kN, 30 kN, 60 kN, 91.2 kN, 60 kN, 30 kN, and 0 kN. The strain and displacement data are collected each time the loading and unloading process is completed.

(2)Fatigue Test Process

The fatigue load amplitude of fatigue test loading is 15 kN → 62.7 kN. The machine is stopped for a static load test when the cycle times reach 50,000, 100,000, 200,000, 500,000, 800,000, 1,000,000, 1,500,000, and 2,000,000. The model is observed during the test process for cracking and other abnormal phenomena, and after each loading, three data acquisitions are completed [29,30,31,32,33].

The loading amplitude is increased accordingly, and data are collected again every 200,000 cycles in the subsequent loading process if the structure does not have fatigue cracking when the fatigue load cycle times reach 2 million. In other words, the machine is stopped for a static load test when the load reaches 2.2 million, 2.4 million, and 2.6 million times, and the strain, displacement, fatigue cracking, and expansion of the measuring points are recorded. When the fatigue load cycle reaches 3 million times, if the structure is still intact, the fatigue loading is stopped.

### 3.4. Test Results and Comparative Analysis

#### 3.4.1. Comparative Analysis

(1)Stress Comparison of Control Measuring Points

Table 1 displays the stress results of measuring points near the diaphragm’s arc opening. Two-way or three-way strain rosettes were used at the junction of the diaphragm and top plate to study the stress distribution of the key parts. The data show the stress results of the diaphragm’s greater side of stress.

Table 2 displays the stress results of U-rib control measuring points. U02, U05, U09, U15, and U22 are all measuring points at the bottom of the U-rib. U11–U13, U16–U18, U19–U21, and U23–U25 are all measuring points at the connection between the U-rib and the top plate. These measuring points are all measured using three-dimensional strain rosettes.

The FEM calculation results of each measuring point of the diaphragm are in good agreement with the measured values, as shown in Table 1. The shoulder on the right side of the 6# rib opening acts as the fixed end of the cantilever beam. The stress at the measuring point is high due to the large bending moment generated here from bearing the vertical load transmitted by the U-rib. The maximum error occurs at the U15 measuring point, as shown in Table 2, and the difference between the theoretically calculated value and the measured value of each U-rib measuring point is smaller within the action range of the separation loader. At the junction between the top plate and the U-rib, the stiffnesses of the diaphragm and the U-rib are quite different because the diaphragm and the U-rib are perpendicular to each other in space. The out-of-plane deformation of the diaphragm has little effect on the stress distribution of the U-rib, so the stress of the U-rib is small here. Because the diaphragm and the U-rib are perpendicular to one another in space at the junction between the top plate and the U-rib, their stiffnesses differ significantly. The diaphragm’s out-of-plane deformation has little impact on the U-rib’s stress distribution, so the stress of the U-rib is small here.

(2)Displacement Comparison of Control Measuring Points

The data of the S01, S02, and S03 vertical displacement measuring points can be obtained through the FEM calculation. The values are 2.68 mm, 2.82 mm, and 2.68 mm, respectively. These values are not significantly different from the test’s average vertical displacement of 2.81 mm, and the maximum error is 4.63%. The vertical displacement of the structure reveals its overall stiffness. The vertical deflection decreases as stiffness increases. The FEM model’s displacement value is relatively close to that of the actual structure, indicating that it has a similar stiffness.

#### 3.4.2. Fatigue Cracking of Experimental Model

The loading amplitude ∆p reaches 47.7 kN when the fatigue loading cycle process reaches 2 million times. A 7.5 mm long crack that is clearly visible can be seen on the upper right side of the 6# opening in Figure 11, and no other parts of the structure show fatigue cracking. As shown in Figure 12, when the fatigue loading cycle reaches 2.2 million times, the corresponding loading amplitude is 1.5*∆p, and the crack extends to 12 mm. As shown in Figure 13, the crack extends to 21 mm when the fatigue loading cycle reaches 2.4 million times. The corresponding loading amplitude is 2*∆p. According to Figure 14, when the fatigue loading cycle reaches 2.6 million times, the corresponding loading amplitude is 2*∆p and the crack reaches a length of 31 mm. The cracks are located in the red squares in Figure 12, Figure 13 and Figure 14 and the corresponding crack lengths have been marked in the corresponding figures. The 159 # and 160 # measuring points are located above and below the cracks, respectively.

## 4. FEM Details of Diaphragm’s Arc Opening Region

### 4.1. Load Case

The longitudinal ribs from right to left are numbered 1# to 7#. The location of concern is the opening area of the right upper diaphragm of the 6# rib where cracking was observed during the fatigue test. The initial position of the unilateral moving load was located directly above the 6# rib. Each time, the moving load was shifted by 25 mm six times. There are a total of seven lateral loading cases (LC1–LC7) that are analyzed.

Through the analysis of the detailed stress responses under seven load cases, it was observed that with the greater distance of the moving loads from the 6# U-rib, the stress level in the details of the diaphragm’s arc opening first gradually increased and then decreased significantly, which suggests the significance of the local effects of the moving loads on the details.

In the following analysis, three lateral load cases closely related to the stress response in the vulnerable area of the diaphragm’s arc opening were selected, as represented in Figure 15. LC1 denotes that the moving load center line on one side was directly above the 6# U-rib. LC4 denotes that the moving load center line on one side was located at the intersection of the web on the right side of the 6# U-rib and the deck top plate, and LC7 denotes that the moving load center line on one side was located at the center of the 6# U-rib and 5# U-rib relative to the web.

After three lateral moving load cases closely related to the detailed stress were selected and analyzed, the moving load moved longitudinally from each lateral position, with a total displacement of 2100 mm. The moving loads at the longitudinal loading position and loading steps are represented in Figure 16. The numbers 1 to 19 in Figure 16 correspond to the Load Steps in Table 3. To determine more accurate stress response results, a step increase was performed near the middle diaphragm (HGB2) and the side diaphragm (HGB3).

The moving load acting directly above the middle diaphragm (HGB2) is considered as the coordinate origin, and the side diaphragm (HGB3) is considered as the positive direction. The specific coordinates are listed in Table 3.

### 4.2. Stress Analysis of Key Points

Based on the statistical data on fatigue diseases of steel bridges, it can be inferred that the location of the most common fatigue cracks around the diaphragm’s arc opening is the location where the radius of curvature of the free edge of the opening is small. This is consistent with the FEM stress results represented in Figure 17. There is a strong stress concentration within a certain range at the intersection of the straight line segment and the arc line segment at the free edge of the opening.

On the basis of this phenomenon, the position of the stress analysis point A is located at the intersection of the straight line segment, and the arc line segment at the free edge of the diaphragm’s arc opening on the right side of the 6# U-rib, and the relative position is demonstrated in Figure 18.

### 4.3. Mesh Independence Check

There is an obvious stress concentration around the opening of the diaphragm plate, and the stress gradient is very large. If the FEM mesh is too large, it may not capture the rapid changes in stress. If the mesh generation is too fine, the overall computing efficiency is impacted [34,35,36,37,38]. Therefore, reasonable mesh generation is required to ensure mesh quality. To obtain stress conditions that are less affected by the mesh generation, three mesh generation schemes, represented in Figure 19, are compared in this paper.

The no. 1 mesh scheme, no. 2 mesh scheme, and no. 3 mesh scheme have element lengths of 3 mm, 2 mm, and 1 mm, respectively. The mesh size along the vertical free edge direction must not be changed for at least five layers to ensure that the vertical and horizontal proportions of the element tend to be 1:1 and the mesh division remains as orthogonal as possible.

At the diaphragm’s arc opening area, the most unfavorably lateral load position is LC4 (the moving load center is located at the junction of the web near the 5# or 6# U-rib and the deck’s top plate). The most unfavorably longitudinal loading position is 0.14 m away from the middle diaphragm, where the load step number is 3.

The stress response of point A under the most unfavorable longitudinal and transverse loading cases is used to check mesh independence. Table 4 shows the nominal stress at 6 mm along the normal opening edge for comparison. Tensile stresses are indicated by positive values in Table 4, while compressive stresses are indicated by negative values. The positive and negative meanings of stress data in this paper are the same as those in Table 4.

With the further refinement of the no. 1 mesh scheme to the no. 2 mesh scheme, the stress data are changed. For the no. 2 mesh scheme and no. 3 mesh scheme, the total stress of the studied details is roughly the same as the in-plane and out-of-plane stress components. Therefore, it can be inferred that when the mesh size is refined to 2 mm or less, the stress level of the studied details no longer fluctuates significantly. Numerical solutions independent of mesh generation can be determined. To facilitate the extraction of the nominal stress and the reference point stress required by the HSS method, the no. 3 mesh generation scheme was selected for FEM analysis.

## 5. Analysis of Details of Opening in Diaphragm’s Arc Opening Area

### 5.1. Stress Extraction Method

There is no consistent definition of the nominal stress extraction method for the diaphragm’s arc opening area of an OSD in the field of steel bridge fatigue research. The existing nominal stress extraction methods for this detail are synthesized in this paper, along with selecting the nominal stress at 2 mm, 6 mm, 10 mm, 13 mm, and 15 mm away from the normal direction of the edge of the stress concentration point. The stress concentration point at the detail can be considered a hot spot through the basic idea of the HSS method. HSS is solved using multipoint linear interpolation and extrapolation. The corresponding fatigue life assessment is carried out with the help of the S–N curve recommended by IIW.

The extrapolation formula of the HSS method adopts the two-point interpolation linear extrapolation from Equation (3), suggested by DET Norske Veritas.

In this paper, the stress peak at point A in this area was used as the reference point for FEM analysis, and the nominal stresses at 2 mm, 6 mm, 10 mm, 13 mm, and 15 mm, respectively, away from point A and the HSS at this point were obtained.

Since the detailed stress responses under the most unfavorable longitudinal and transverse load cases are analyzed, only the nominal stress and HSS at all locations under LC4 load cases were extracted. For LC1 and LC7, only nominal stress and HSS at 6 mm were extracted for comparison. The schematic diagram of nominal stress and HSS extraction is represented in Figure 20. These two evaluation methods do not consider the influence of nonlinear stress factors.

### 5.2. Stress Results Analysis of Arc Opening

Within a certain range of moving load, the second and third principal stresses near the peak point A of the diaphragm’s arc opening stress are negative, which can be analyzed from the FEM results. This suggests that point A is under biaxial compression within a certain range. Considering the third principal stress as the representative to evaluate the fatigue performance, the large absolute value of the third principal stress at nearby points can be determined. This also meets the requirements of stress-extracting parameters by using the nominal stress method and HSS method.

There is a marginal difference in the stress on the front and back of the stress peak at point A, due to the thickness of the plate. Stress extraction was carried out on the side with a larger value from the perspective of conservative fatigue assessment. Figure 21 represents the variation curves of nominal stress and HSS at stress peak at point A under three load cases.

The stress level under LC4 is the highest for the nominal stress value and HSS value at 6 mm, and the stress level under LC7 is the lowest during the longitudinal and transverse loading of the entire moving loads, as shown in Figure 21.

LC4 is the most unfavorable lateral loading position. The moving load center line is parallel to the connection between the web plate and the top plate at the side of the 6# U-rib and 5# U-rib. The position with the most unfavorable longitudinal loading is 140 mm away from the middle diaphragm plate (HGB2). The overall change in stress indicates an increasing trend at first, followed by a gradual decrease.

When the moving load is directly above the side diaphragm (HGB3), the principal stress at point A is a mere −2.54 MPa (considering the nominal stress at 6 mm under LC4). Compared with the most unfavorable loading case, the principal stress is reduced by 30.74 MPa, accounting for 7.64% of the stress.

When the moving load is 750 mm away from the right side of the diaphragm (HGB3), the principal stress is only −0.78 MPa. It can be inferred that only when the moving load acts within a certain range do the studied details have a better stress response.

The stress influence line of point A along the longitudinal bridge direction is roughly distributed in a triangular shape. It can be concluded from the stress response results that the length of the stress influence line is almost twice the total length of the longitudinal movement, which is 4.2 m. If the axle load in the fatigue vehicle is applied at the peak point of the stress influence line, the front and rear axles must be applied within 2.1 m on the left and right of the middle diaphragm to have a significant impact on the stress results, according to the arrangement of the most unfavorable loading scheme.

The influence of the front and rear axles on the stress response is not considered in this paper, and it is appropriate to only consider the middle axle for simulated loading. Under the three loading cases, the variation curves of the third principal stress at the stress peak point with the longitudinal moving load position are represented in Figure 21. It can be analyzed from the stress distribution that the nominal stress and the HSS derived from two points have similar change trends. The HSS method is more conservative than the nominal stress method.

The values of the principal stresses under LC4 are mentioned in Table 5. The nominal stress values reduce significantly with the increase in the normal distance of the opening edge, suggesting that the stress gradient at this point is large.

### 5.3. Displacement Results Analysis

Under the three lateral load cases in Section 4.1, the displacement and deformation of the middle diaphragm (HGB2) under longitudinal load steps i = 2 (LC1) and i = 3 (LC4 and LC7) are represented in Figure 22. When the moving load is exactly above the diaphragm, the diaphragm only produces in-plane deformation, and there is no out-of-plane bending stress on the diaphragm web, the in-plane membrane stress takes a major portion.

Figure 22 represents that there is a strong bending shear effect between the U-rib and the diaphragm in the plane. Some parts of the axial force and shear force in the diaphragm are transmitted through the longitudinal ribs. The transverse diaphragm and longitudinal rib affect each other. The middle part of the diaphragm is subject to the most significant compression, so the bending moment is the largest.

The closer to the end of the diaphragm, the more impactful the shear force is. The longitudinal ribs are twisted, leading to different stress signs on both sides of the longitudinal rib. Under the comprehensive influence of bending and shearing of the longitudinal rib and diaphragm, significant stress concentration occurs at the end of the weld and arc opening.

Simultaneously, Figure 22a indicates that when the load is only acting on the top of the 6# U-rib, as the diaphragm is drilled to release the secondary bending stress at the welding between the diaphragm and the U-rib, the webs of the U-rib and the diaphragm are staggered in the vertical direction and squeezed in the horizontal direction, resulting in tensile stress at the two welds.

Figure 22b,c show that when the moving load deviates from the center line of the U-rib for loading, the U-rib is distorted, and its bottom generates lateral displacement. Furthermore, the diaphragm serves as an embedded boundary to limit displacement. The discontinuous embedded boundary around the U-ribs, on the other hand, causes significant out-of-plane deformation of the U-ribs, and the weld end and opening periphery are subject to higher secondary stress.

### 5.4. Stress Composition of Diaphragm’s Arc Opening Area

The moving load with an area of 305 mm × 175 mm acts directly above HGB2. The initial position is at the central axis of the 6# U-rib, and it then moves horizontally to the right with an amplitude of 25 mm each time. During the movement, point A with obvious stress concentration can be considered as the representative to determine the stress composition and change law in the diaphragm’s arc opening area.

The front side denotes the side with larger value where the peak stress point A of the middle diaphragm plate is closer to the moving load action position, and the back side denotes the side with a smaller value, away from the moving load action position.

Under the LC1, LC4, and LC7 load cases, the nominal principal stress at 6 mm from the normal direction of the edge of the arc opening at point A is analyzed. If the principal stress directions of the front and back sides of point A are roughly the same, it can be determined that $\sigma_{front}$ is obtained with the linear addition of in-plane stress and out-of-plane stress, and $\sigma_{back}$ denotes the difference between in-plane stress and out-of-plane stress. On this basis, the stress composition and variation law of point A are studied.
(15)$$\left\{\begin{array}{l} \sigma_{in-plane}=\frac{\sigma_{front}+\sigma_{back}}{2}\\ \sigma_{out\;of\;plane}=\frac{\sigma_{front}-\sigma_{back}}{2} \end{array}\right.$$

$\sigma_{front}$ denotes the stress value at the side near the moving load’s action position at point A of the diaphragm plate, and $\sigma_{back}$ denotes the stress value at the side away from the moving load’s action position at point A of the diaphragm plate.

If the principal stress directions of the two sides are different, they cannot be treated using the above methods but should be decomposed into normal stress and shear stress to conduct fatigue performance analysis.

It can be observed from Figure 23a,c,e that the stress values on the front and back of the stress peak at point A are close. From the subsequent FEM calculation, it can be observed that the principal stress direction is basically the same, so the fatigue performance can be analyzed through the linear decomposition of the principal stress.

Figure 23a suggests that the stress at point A is negative, indicating that the point is under pressure. When the load moves to the right, it can be observed that the stress value changes significantly during the initial movement, with an increase of 2.53%. This indicates that the stress at the point of concern is closely related to the lateral action position of the moving load.

When the moving load shifts 75 mm to the right, the stress reaches the maximum value of −45.40 MPa, with an increase of 4.30%. At this time, the center of the moving load on one side coincides with the connection between the web of the 6# U-rib, which is near the 5# U-rib, and the top plate of the deck. Further, the moving load continues to move to the right three times (25 mm each time, 75 mm in total). It can be observed that the stress reduces significantly during the moving load movement (the decreases are 5.07%, 5.14%, and 8.36%, respectively), and eventually acts on the center line of the 6# U-rib and 5# U-rib relative to the web. The change in the moving load’s lateral action position has a significant local effect on its stress response. The local effect is related to the force transmitted by the weld between the U-rib and the diaphragm. The closer the moving load center is to the web of the U-rib on the side of point A, the greater the force transmitted by the weld, resulting in an increase in the stress level at point A. To sum up, LC4 is the most unfavorable load case in the transverse direction, and LC7 is the case with the minimum load effect.

As depicted in Figure 23b, when the moving load moves laterally on the middle diaphragm, the out-of-plane stress of point A is 0, and the in-plane stress is dominant. Since the stiffening rib of the diaphragm is set at one side, the diaphragm structure is not completely symmetrical, so there is a small amount of out-of-plane stress. Starting from the corresponding transverse loading positions of LC1, LC4, and LC7, it moves 2100 mm longitudinally towards the side diaphragm (HGB3).

As highlighted in Figure 23c,e,g, when the moving load moves longitudinally, the front stress (near the side of the moving load) of point A under all load cases is marginally greater than the back stress. When the center line of the load’s resultant force is 140 mm away from the middle diaphragm, the stresses on both sides of point A (under the LC4 load case) attain the highest level simultaneously. The main reason for this phenomenon is that when the load is directly above HGB2, the diaphragm plays a leading role in lateral diffusion, which decreases the in-plane stress level at point A. When the load deviates from HGB2 by a certain distance, the top plate is not supported by the diaphragm, leading to its low bending stiffness. The load effect is primarily transmitted to the weld between the U-ribs and the diaphragm longitudinally through the U-ribs, which enhances the stress concentration at point A.

It can be observed from the in-plane and out-of-plane stress distribution in Figure 23d,f,h that under the three load cases, the maximum combined effect of in-plane and out-of-plane stress is located at the most unfavorable longitudinal loading position (load steps i = 2, i = 3, and i = 3, respectively). However, the proportion of out-of-plane stress can be ignored, as it is 0.01% under LC1, 0.26% under LC4, and 0.32% under LC7. When the moving load center acts in the middle of HGB2 and HGB3, the vertical downward deflection of the U-rib is the largest. At this time, the out-of-plane deformation and stress of the diaphragm web plate are at the maximum. The maximum out-of-plane stresses under the LC1, LC4, and LC7 load cases are −0.43 MPa, −0.44 MPa, and −0.40 MPa, respectively.

Under the LC1 load case, the proportion of out-of-plane stress is 1.46%, which is still negligible. Although the out-of-plane stress simultaneously reaches the peak, the in-plane and out-of-plane combined effect has been reduced to 67.9% under the most unfavorable loading. Under the LC4 load case, the out-of-plane stress accounts for 1.36%, and the in-plane and out-of-plane combined effect is reduced to 73.15% under the most unfavorable loading. In the case of LC7, the out-of-plane stress accounts for 1.32%, and the in-plane and out-of-plane combined effect is reduced to 78.19% under the most unfavorable loading.

In conclusion, the contribution of out-of-plane bending stress to the overall load effect is very small. Irrespective of the changes in the moving load position, the load effect in this area is primarily caused by in-plane stress.

Regardless of the influence of material properties, stress amplitude is the key factor in the fatigue evaluation of steel bridges. Therefore, it can be considered that the chief cause of fatigue cracking at the peak point A of a diaphragm’s arc opening is in-plane stress [30,39,40,41,42,43]. Out-of-plane stress has a minor effect on fatigue cracking at the peak point A.

### 5.5. Control Stress of Arc Opening’s Cracking

Theoretical computation value derivation was carried out for the longitudinal load step i = 3 under the LC4 load case. The six stress components of point A under the LC4 load case were extracted. The specific values are as follows:(16)$$\left\{\begin{array}{c} \sigma_{x}=-4.30\\ \sigma_{y}=-43.46\\ \sigma_{z}=-0.0012\\ \tau_{\mathrm{xy}}=-8.21\\ \tau_{\mathrm{yz}}=0.095\\ \tau_{\mathrm{zx}}=0.039 \end{array}\right.$$

By calculation:(17)$$\left\{\begin{array}{c} \varphi_{1}=\sigma_{x}+\sigma_{y}+\sigma_{z}=-47.7612\\ \varphi_{2}=\sigma_{x}\sigma_{y}+\sigma_{y}\sigma_{z}+\sigma_{x}\sigma_{z}-\tau_{\mathrm{xy}}^{2}-\tau_{\mathrm{yz}}^{2}-\tau_{\mathrm{zx}}^{2}=119.52\\ \varphi_{3}=\sigma_{x}\sigma_{y}\sigma_{z}-\sigma_{x}\tau_{\mathrm{yz}}^{2}-\sigma_{y}\tau_{\mathrm{zx}}^{2}-\sigma_{z}\tau_{\mathrm{xy}}^{2}+2\tau_{\mathrm{xy}}\tau_{\mathrm{yz}}\tau_{\mathrm{zx}}=-0.0993 \end{array}\right.$$

Substituting (17) into Equation (8):$$\sigma^{3}+47.7612\sigma^{2}+119.52\sigma +0.0993=0$$

Then:(18)$$\left\{\begin{array}{c} \sigma_{1}=-0.00083\\ \sigma_{2}=-2.65\\ \sigma_{3}=-45.11 \end{array}\right.$$

Substituting (16) and (18) into Equation (11):(19)$$\left\{\begin{array}{c} A_{3}=\tau_{xy}\tau_{yz}-\left(\sigma_{y}-\sigma_{3}\right)\tau_{zx}=-0.8443\\ B_{3}=\tau_{xy}\tau_{zx}-\left(\sigma_{x}-\sigma_{3}\right)\tau_{yz}=-4.1971\\ C_{3}=\left(\sigma_{x}-\sigma_{3}\right)\left(\sigma_{y}-\sigma_{3}\right)-\tau_{xy}^{2}=-0.0676 \end{array}\right.$$

Substituting (19) into Equation (10):(20)$$\left\{\begin{array}{c} l_{3}=\frac{A_{3}}{\sqrt{A_{3}^{2}+B_{3}^{2}+C_{3}^{2}}}=-0.197\\ m_{3}=\frac{B_{3}}{\sqrt{A_{3}^{2}+B_{3}^{2}+C_{3}^{2}}}=-0.98\\ n_{3}=\frac{C_{3}}{\sqrt{A_{3}^{2}+B_{3}^{2}+C_{3}^{2}}}=-0.015 \end{array}\right.$$

The angle between the maximum absolute principal stress and the X-axis is 101.37°, which represents the angle between the normal direction of the principal stress and the X-axis, which is 11.37°. The principal stress direction is 90.86° to the Z-axis of the longitudinal bridge, which confirms that the diaphragm’s arc opening area is primarily under in-plane stress.

For the most unfavorable longitudinal and transverse loading, the second and third principal stress directions around the diaphragm’s arc opening are highlighted in Figure 24a. The third principal stress is parallel to the edge tangent. Within a certain distance from the edge normal, the direction of the third principal stress still maintains the original angle. The crack is located in the red square in Figure 24b, and the crack length have been marked in the corresponding figure. The 159 # and 160 # measuring points are located above and below the crack, respectively.

The theoretical calculation results show that the normal direction of the principal stress has an included angle with a X-axis of 11.37°. The normal direction of the third principal stress at point A is 11.1° from the horizontal plane, according to the principal stress direction field computed using the FEM calculation results. This is similar to the fatigue test’s crack propagation angle of 12.8°. As shown in Figure 25, the change curve of the third principal stress along the X-axis of the extraction path under the most unfavorable loading is drawn with the normal direction of the third principal stress as the X-axis direction.

It can be observed from the above figures that the third principal stress changes significantly along the normal direction of the opening edge. When the distance to the extracted nominal stress is more than 6 mm, the stress value reduces linearly, and the stress concentration effect decreases obviously. This suggests that the nominal stress is significantly affected by the location of stress extraction. Fatigue assessment based on the HSS method leads to relatively conservative results. The stress value is relatively stable and does not fluctuate significantly with the location of stress extraction.

The stress responses for the stress distribution at the free edge of the opening when the longitudinal load step i = 3 are shown in Figure 26. The curved value of the extraction path is represented by the abscissa t in Figure 26. Because of the significant stress concentration at the opening, the origin of the abscissa is set 10 mm ahead of point A.

In all three key load cases, the free edge of the diaphragm’s arc opening is under pressure. The peak stress occurs at t = 10 mm (point A) and at the symmetrical position on the other side. The stress decreases gradually with the increase in the thickness values, and the stress near the middle of the hole is almost zero. The region near the hot spot is in a biaxial compression state. The premise of fatigue crack growth is that the region is in a “pure tension” state. It is speculated that the total stress in this area may vary from compression to tension due to the influence of initial defects and welding residual tensile stress. Eventually, the cyclic action of higher stress amplitude results in the occurrence of fatigue cracking.

The calculation results state that the fatigue crack control stress at the peak point A of the diaphragm’s arc opening stress is the principal stress with a maximum absolute value, and the stress direction is tangential to the hole edge. Although the area is primarily compressed under the load, the residual tensile stress at the connection between the U-rib and the diaphragm causes it to have similar fatigue characteristics to the weld details. The reciprocating compressive stress around the diaphragm’s arc opening can also cause fatigue cracking.

## 6. Fatigue Performance Evaluation of Diaphragm’s Arc Opening

### 6.1. Assessment of Fatigue Effects in Open Hole Area

The stress response of the diaphragm’s arc opening stress peak at point A under different load cases was extracted under the premise of the similarity in stress distribution under lateral loading cases. The stress cycle curves are depicted in Figure 27. The stress states of the diaphragm’s arc opening stress peak at point A under the most unfavorable load case LC4 are analyzed. It can be noted from Figure 27b that the diaphragm primarily bears in-plane membrane stress, as is known from the previous conclusion. Vertical stress $\sigma_{y}$ has the most obvious variation, and its stress level is significantly greater than those of the other stress components. Vertical stress $\sigma_{y}$ accounts for the largest proportion of fatigue cracking at point A. The fatigue crack in the model test has an angle of 12.8° included with its normal direction.

The chief reason that the cracking direction is not perpendicular to its stress direction is the existence of the other stress components. The transverse normal stress $\sigma_{x}$ is higher than other stress components, which promotes further propagation of the crack along the transverse direction. The local impact of the wheel makes the level of shear stress higher, which is the second in all stress components.

Moreover, the residual stress component is almost negligible. From the results of the stress cycle curves, it can be inferred that the chief stress components of point A are $\sigma_{x}$, $\sigma_{y}$, and $\tau_{xy}$. These three stress components belong to the stress in the XY plane, suggesting that the region is in a biaxial stress state. The normal stress $\sigma_{x}$ of point A accounts for 10.08% of $\sigma_{y}$, and the shear stress $\tau_{xy}$ accounts for 19.24% of $\sigma_{y}$ when the longitudinal and transverse directions are the most unfavorable. When the load acts directly above HGB2, the normal stress $\sigma_{x}$ of point A accounts for 10.02% of $\sigma_{y}$, and the shear stress $\tau_{xy}$ accounts for 19.13% of $\sigma_{y}$.

It can be observed from Figure 27a–c that the change trend of the third principal stress with the largest absolute value is consistent with that of the vertical normal stress. The stress cycle curve of $\sigma_{m}$ is enveloped by the stress cycle curve of $\sigma_{y}$, suggesting that it is appropriate to use the third principal compressive stress to analyze the fatigue performance of this region.

To deal with the complex biaxial stress state of the structure in a practical structure, this study developed a set of biaxial fatigue effect evaluation methods based on the relevant contents in Section 2.2. The magnitude difference between the primary stress component and the principal stress was used to assess the spatial fatigue effect of the studied details. The parameters defined in the preceding expression describe the degree of deviation between the chief stress component and the principal stress (21).
(21)$$\delta =\frac{\Delta_{\sigma_{n}}}{\Delta_{\sigma_{m}}}$$

$\Delta_{\sigma_{n}}$ denotes the maximum value of the main stress amplitude at all levels.

$\Delta_{\sigma_{m}}$ denotes the maximum value of the principal stress amplitude with the maximum absolute value at all levels.

The calculation results of stress amplitude deviation δ are listed in Table 6.

Under the three load cases, the deviation value, δ, of the stress amplitude first decreases and then gradually increases, as shown in Figure 28, which is roughly consistent with the change trend of the third principal stress at the peak point A of the diaphragm’s arc opening stress. When the moving load is a certain distance away from the diaphragm, the stress amplitude deviation δ is the smallest and the deviation degree of $\sigma_{m}$ and $\sigma_{y}$ is the greatest. The biaxial stress state in the diaphragm is the most significant at this point.

For the third load step, the stress amplitude deviation, δ, gradually increases with the increase in the distance of the moving load from the center line of the 6# U-rib. $\delta_{1}=0.9536$ under LC1, $\delta_{4}=0.9553$ under LC4, and $\delta_{7}=0.9570$ under LC7. This phenomenon suggests that the more the moving load center line deviates from the structural details concerned in the transverse direction, the weaker the in-plane biaxial fatigue effect is. The in-plane biaxial fatigue effect is more significant when the moving load on one side is directly above the 6# U-rib.

### 6.2. Fatigue Life Analysis Based on S–N Curve

When the HSS method is used, IIW recommends using FAT100 and FAT90 fatigue life curves for fatigue life assessment. Since the welding defects do not affect the structural details, the area around the diaphragm’s arc opening must be evaluated with a higher fatigue level. Thus, FAT100 and FAT112 were selected to evaluate the fatigue life of this opening area. The improved luffing fatigue assessment S–N curve is used because the actual situation involves variant loading, as illustrated in Figure 29.

The HSS at point A under the most unfavorable loading is −44.70 MPa, which is greater than the cut-off stress amplitude corresponding to the two S–N curves shown in Figure 29. This suggests that the structural details have a limited life. The corresponding fatigue life is computed using the following equations:(22)$$N=C/\Delta_{\sigma }{}^{m}$$
(23)$$N=\left(365\right)\left(Y\right)n{\left(ADTT\right)}_{SL}$$

*Y* denotes the design life of the construction details. ${\left(ADTT\right)}_{SL}$ is the daily average traffic volume of a single lane. *N* denotes the number of stress cycles caused by one fatigue test vehicle. In this section, ${\left(ADTT\right)}_{SL}$ is taken as 5000, and *N* is conservatively taken as 3. By substituting relevant variables into Equations (22) and (23), the fatigue life assessment results of the studied details can be determined, as mentioned in Table 7.

## 7. Conclusions

This theoretical model was designed based on the structural design parameters of the actual bridge, and the stress responses under different stress extraction methods were compared. The HSS method, which is the chief evaluation method in follow-up fatigue performance research, was used to deal with fatigue cracking in the opening area of the diaphragm during the fatigue test. The control stress of the mentioned category of a fatigue crack was determined by comparing the direction of the principal stress field with that of the crack in the model test. The reasons for fatigue cracking were analyzed, the in-plane fatigue effect of the diaphragm’s arc opening area was evaluated, and the fatigue life analysis was carried out based on the S–N curve. The conclusions are the following.

The stress influence line of the diaphragm’s arc opening area in the longitudinal direction is nearly twice the distance between adjacent diaphragms. In FEM analysis, only the middle or rear axles of the fatigue test vehicle can be loaded. When a fatigue test vehicle passes through the area, this can result in two to three stress cycles.The stress in the diaphragm’s arc opening area is primarily in-plane, with little out-of-plane stress. To deal with the significant stress concentration at the arc opening location, the HSS method can extract stable HSS estimates. The most unfavorable lateral action position of the moving load is the connection between the web plate and the deck roof on the side of the U-rib closest to the study details. When the moving load longitudinally deviates 140 mm (1:2 scale) from the diaphragm plate, the in-plane and out-of-plane combined stress values reach their maximum.Under biaxial compression, the fatigue crack control stress in the diaphragm’s arc opening area is the third principal stress tangential to the hole edge, and is also the principal stress with the largest absolute value. The normal direction of the principal stress is 11.1° with the horizontal line in the X-axis direction, which is approximately consistent with the fatigue test crack initiation angle of 12.8°. Based on the third principal stress, it is possible to conduct a fatigue assessment for this research detail.The stress around the arc opening is negative, and the stress near the opening’s center is near zero. Despite the fact that the area is primarily under pressure, residual tensile stress caused by the welding of the U-rib and diaphragm results in fatigue characteristics with weld details. Therefore, fatigue cracking occurs as a result of the reciprocating compressive stress in the diaphragm’s arc opening area.The cyclic curve of each stress component of the arc opening was studied in detail. The stress level of vertical normal stress $\sigma_{y}$ is considerably higher than that of the other stress components, which accounts for the largest proportion of fatigue cracking. The stress amplitude deviation value δ is used to determine the deviation degree between the stress component $\sigma_{y}$ and the principal stress $\sigma_{m}$ used to evaluate fatigue. δ first decreases and then gradually increases in the longitudinal direction. This indicates that when the moving load is a certain distance from the diaphragm, the deviation degrees of $\sigma_{m}$ and $\sigma_{y}$ are the largest.

## Figures and Tables

**Figure 1 materials-16-05217-f001:**
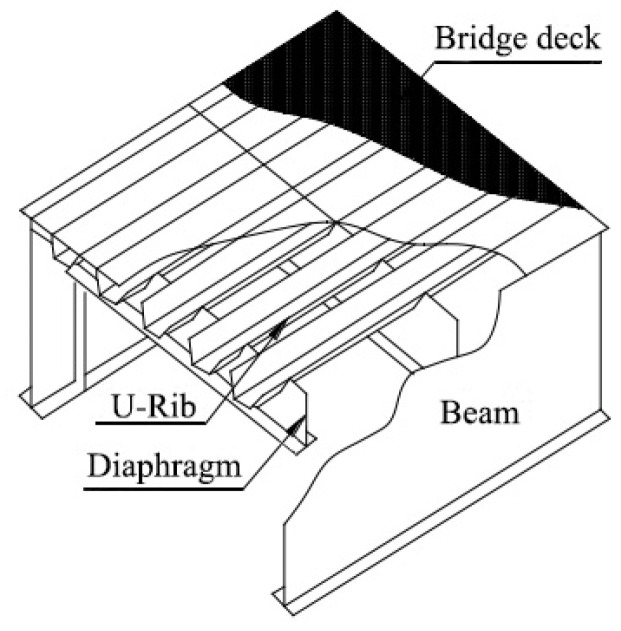
The components of OSD.

**Figure 2 materials-16-05217-f002:**
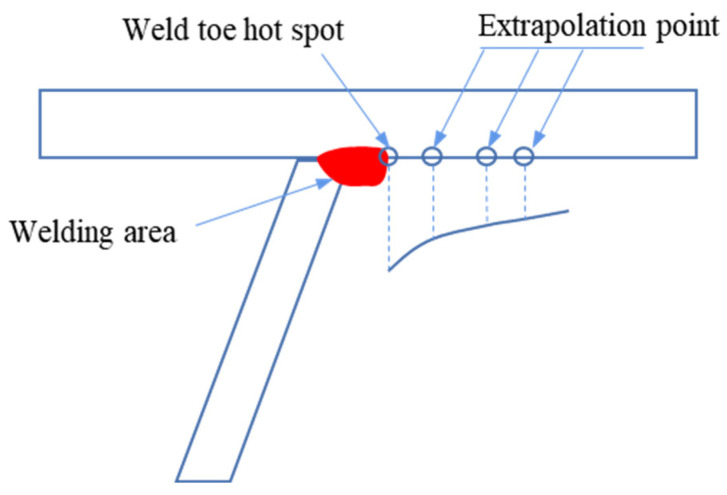
Surface stress extrapolation method.

**Figure 3 materials-16-05217-f003:**
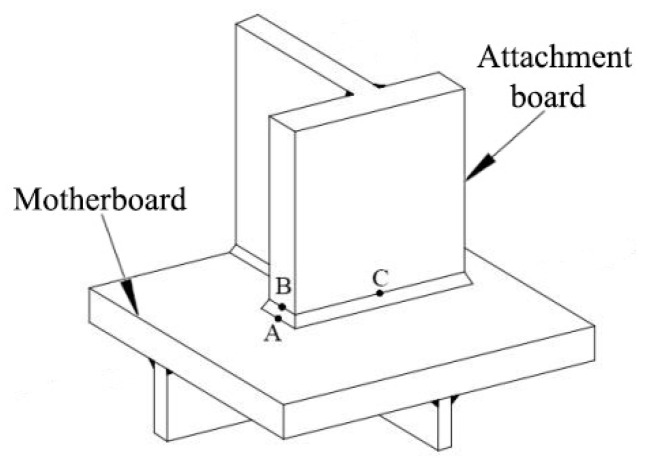
Hot spot types.

**Figure 4 materials-16-05217-f004:**
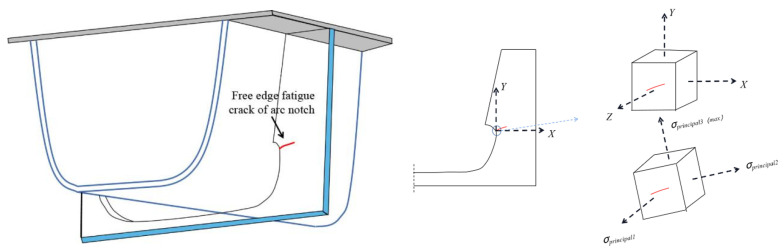
Stress state at the curved openings of diaphragm.

**Figure 5 materials-16-05217-f005:**
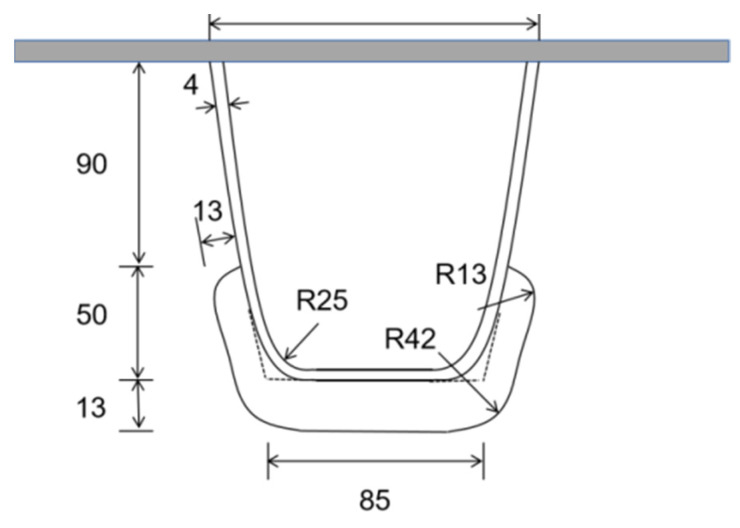
Schematic diagram of U-rib and opening size (mm).

**Figure 6 materials-16-05217-f006:**
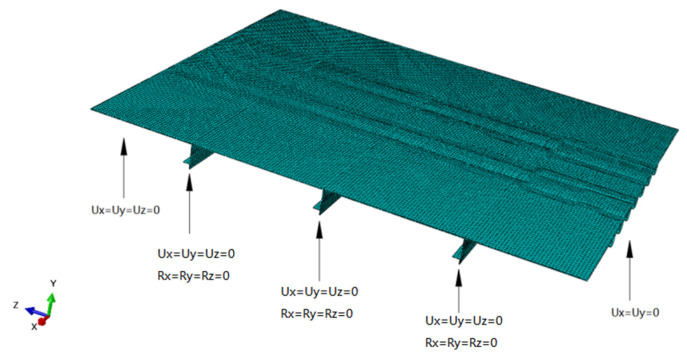
FEM model.

**Figure 7 materials-16-05217-f007:**
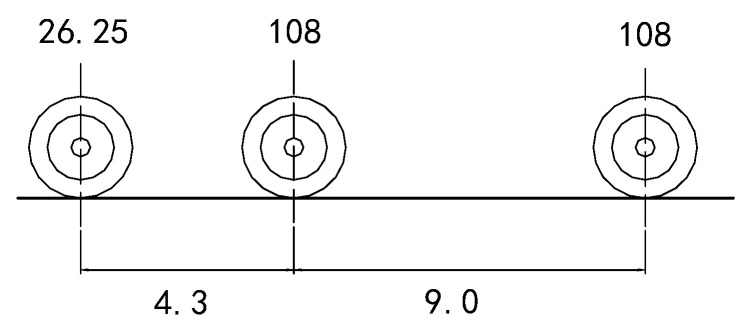
Simplified standard fatigue vehicle HS15.

**Figure 8 materials-16-05217-f008:**
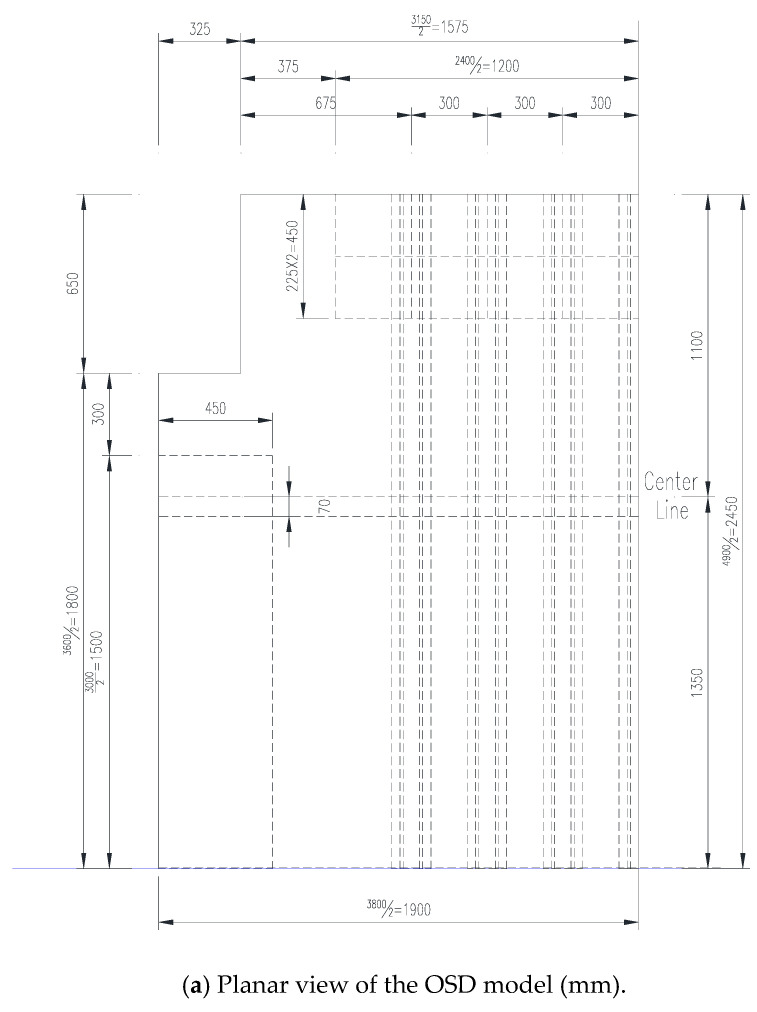
Experimental model.

**Figure 9 materials-16-05217-f009:**
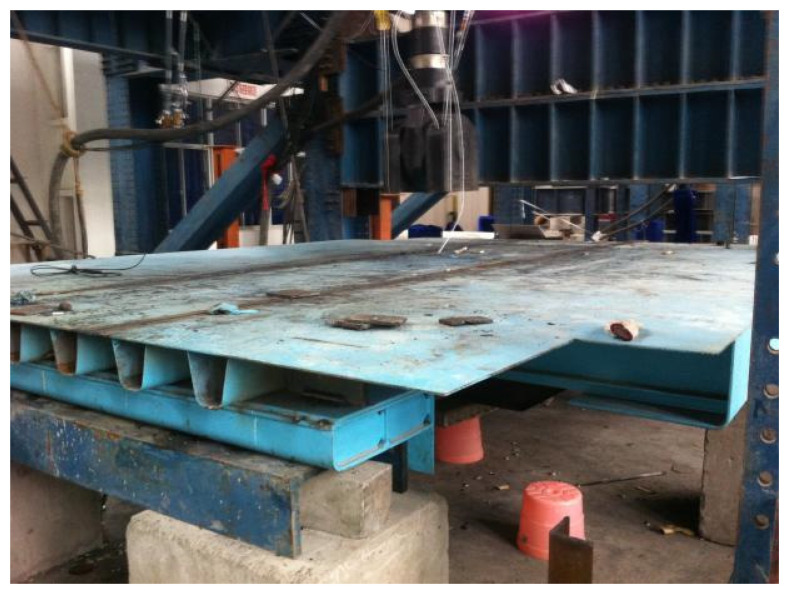
In situ installation of experiment model.

**Figure 10 materials-16-05217-f010:**
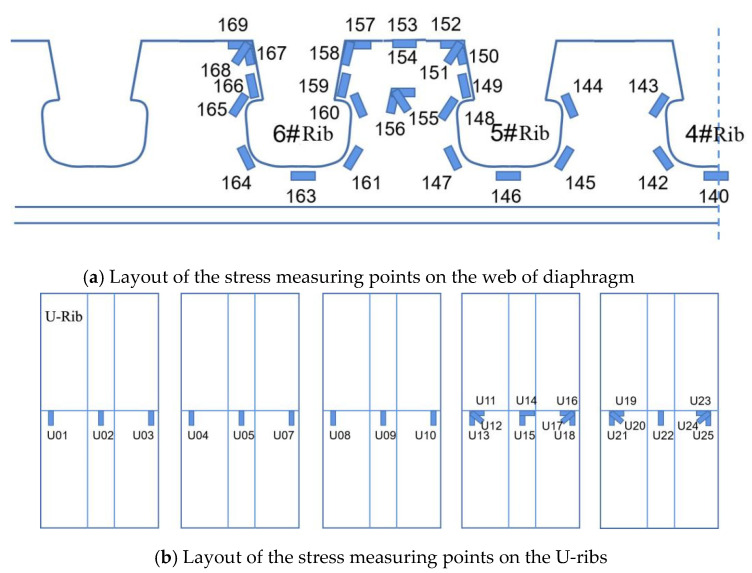
Layout of measuring points on diaphragm and U-ribs: The symbol ‘#’ is used to indicate the numerical order in this paper. E.g., the 6th Rib is the 6#Rib.

**Figure 11 materials-16-05217-f011:**
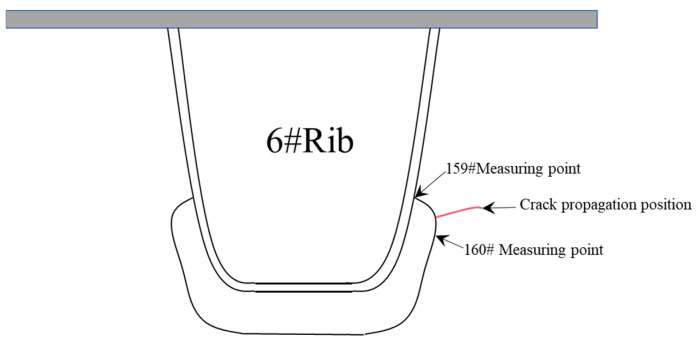
Schematic diagram of fatigue crack’s location.

**Figure 12 materials-16-05217-f012:**
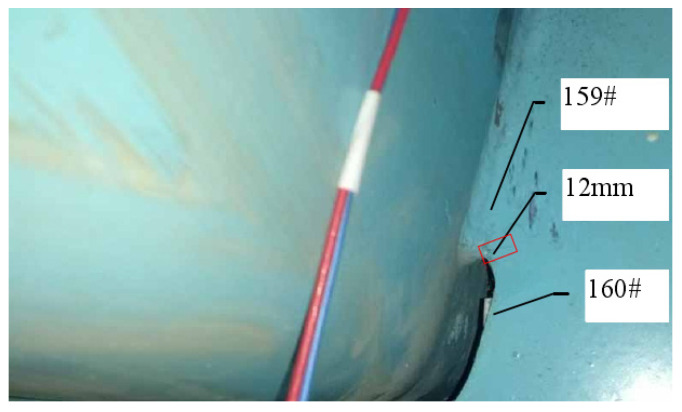
Fatigue cracking at 2.2 million cycles.

**Figure 13 materials-16-05217-f013:**
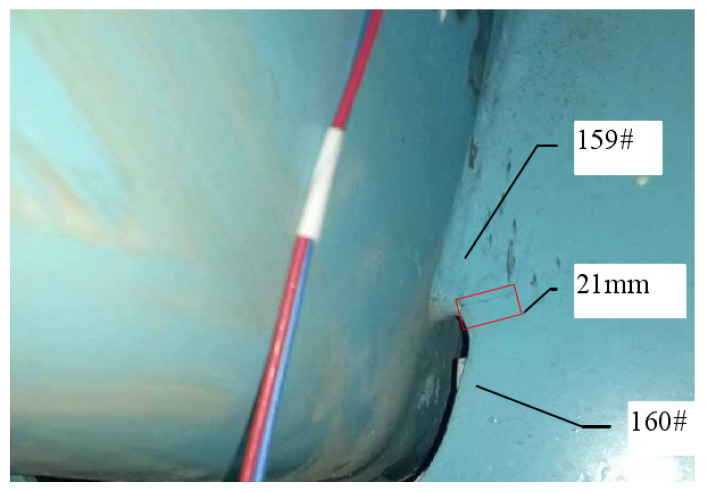
Fatigue cracking at 2.4 million cycles.

**Figure 14 materials-16-05217-f014:**
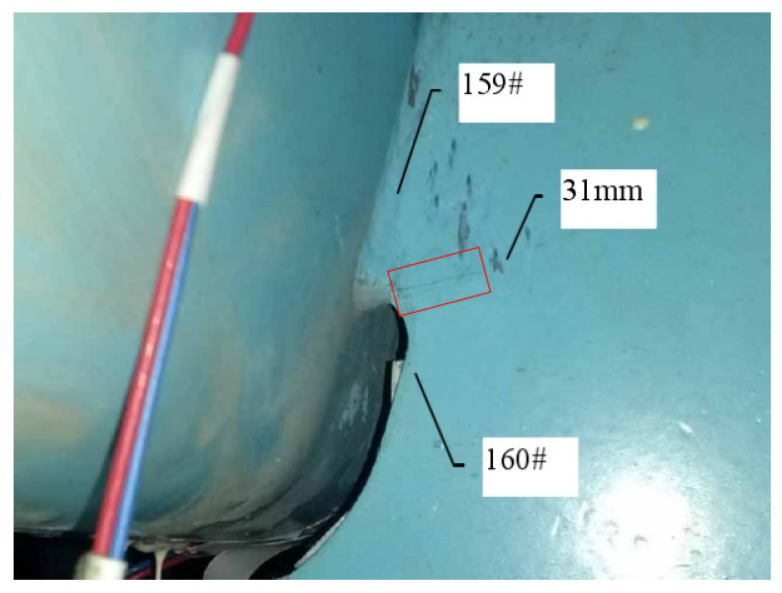
Fatigue cracking at 2.6 million cycles.

**Figure 15 materials-16-05217-f015:**
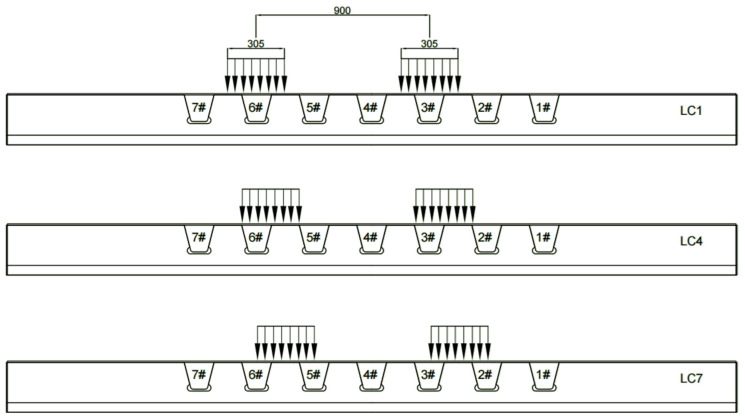
Three mainly analyzed lateral movement load cases (mm).

**Figure 16 materials-16-05217-f016:**
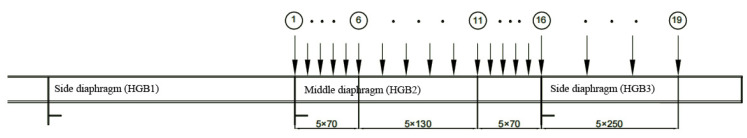
Moving load at the longitudinal movement loading position (mm).

**Figure 17 materials-16-05217-f017:**
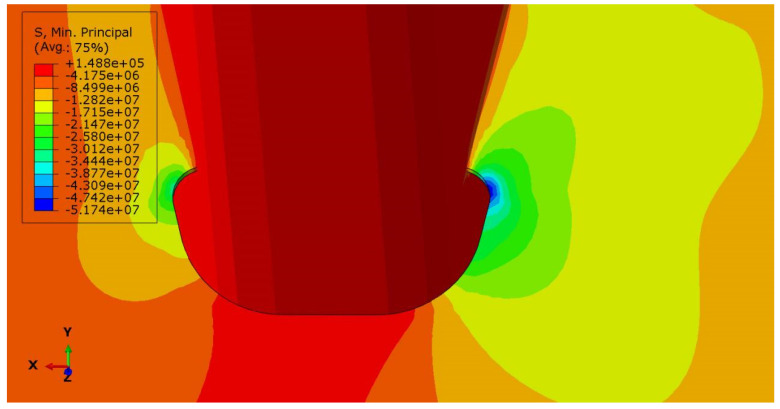
Stress nephogram around diaphragm’s arc opening.

**Figure 18 materials-16-05217-f018:**
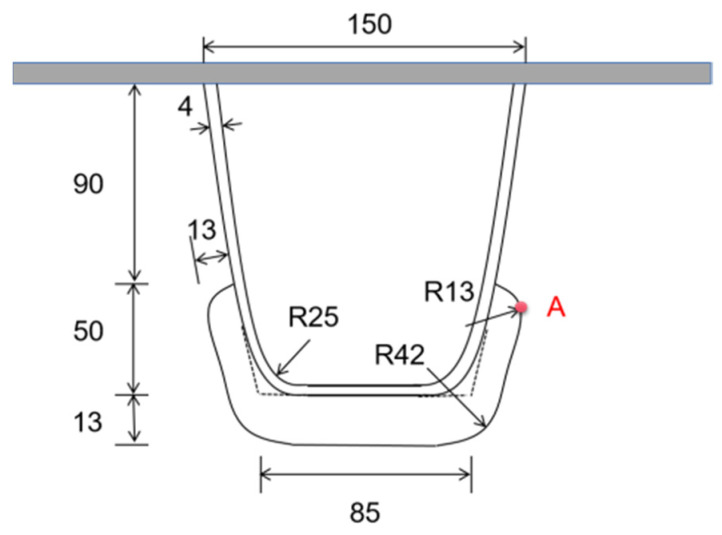
Relative position of stress analysis point A.

**Figure 19 materials-16-05217-f019:**
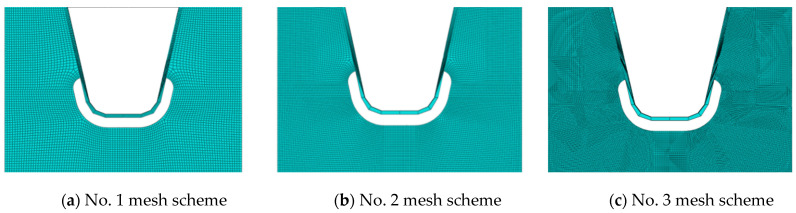
Mesh generation schemes around diaphragm’s arc opening.

**Figure 20 materials-16-05217-f020:**
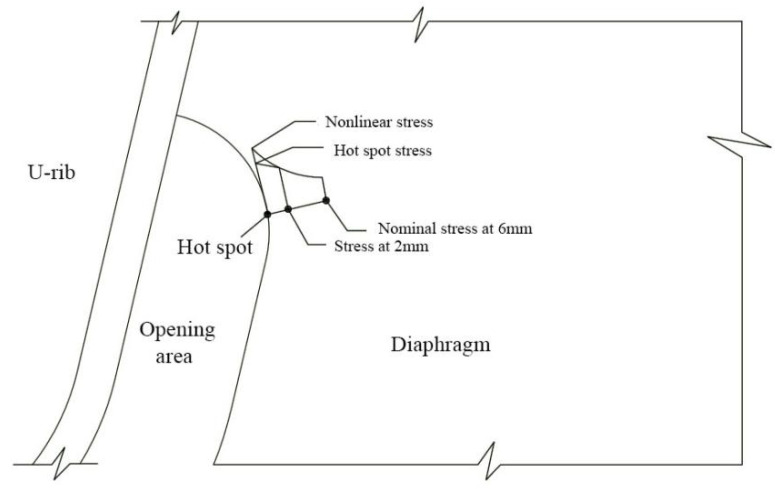
Schematic diagram of stress extraction.

**Figure 21 materials-16-05217-f021:**
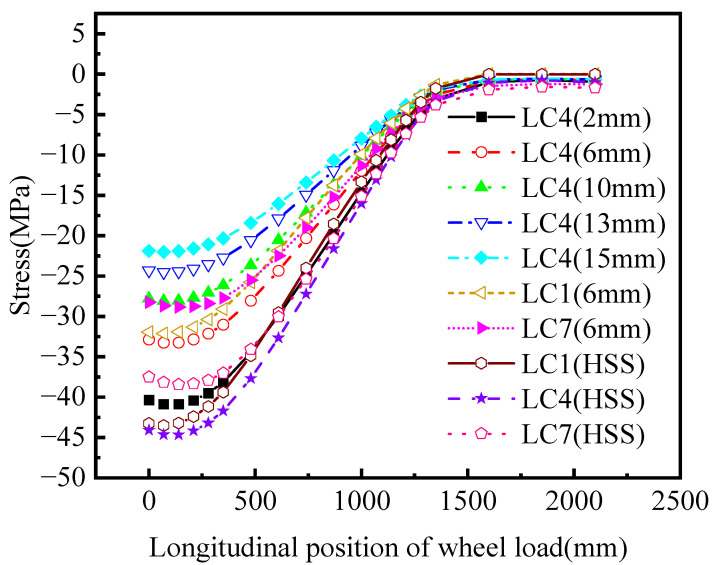
Longitudinal positions of principal stresses and moving loads.

**Figure 22 materials-16-05217-f022:**
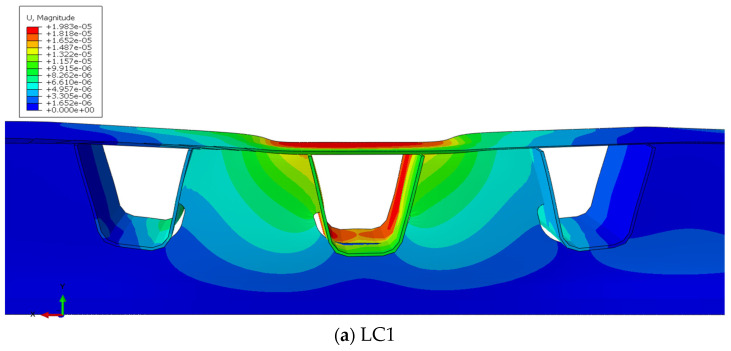
Displacement deformation under the most unfavorable longitudinal load.

**Figure 23 materials-16-05217-f023:**
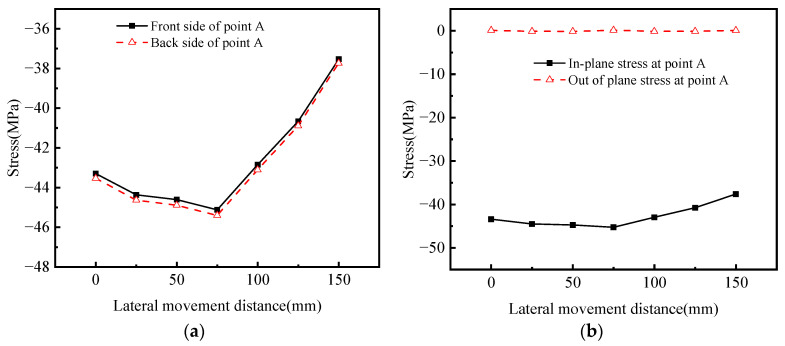
Stress values at the point of interest under longitudinal movement of moving load. (**a**) Stress values of front and back sides when load moves laterally; (**b**) in-plane and out-of-plane stresses during load traverse; (**c**) stress values of front and back sides when load moves longitudinally under LC1 load case; (**d**) stress values inside and outside the plane when the load moves longitudinally under LC1 load case; (**e**) stress values of front and back sides when load moves longitudinally under LC4 load case; (**f**) stress values inside and outside the plane when the load moves longitudinally under LC4 load case; (**g**) stress values of front and back sides when load moves longitudinally under LC7 load case; (**h**) stress values inside and outside the plane when the load moves longitudinally under LC7 load case.

**Figure 24 materials-16-05217-f024:**
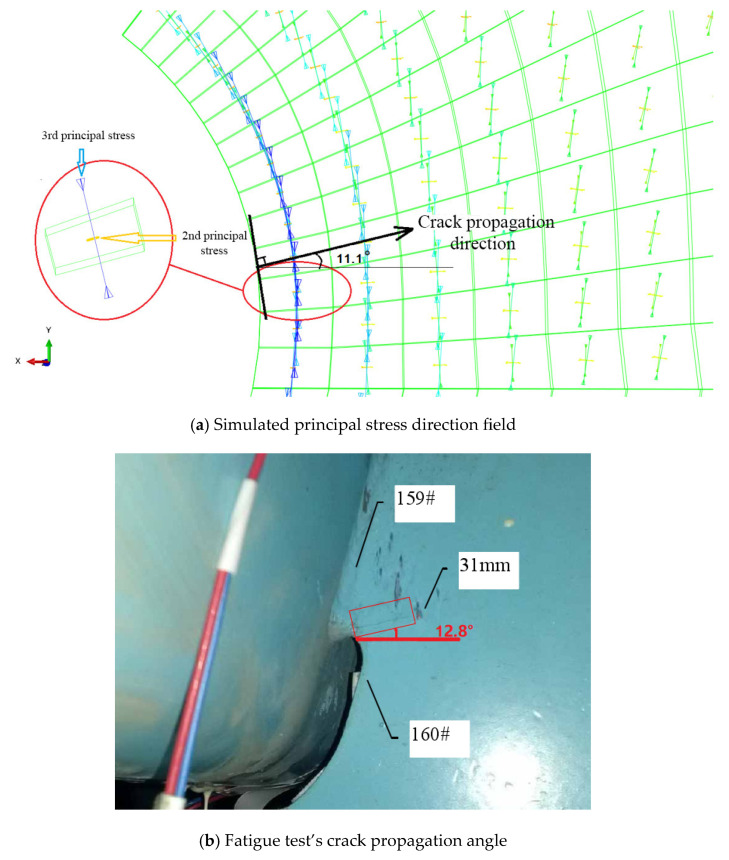
Principal stress’s direction field around the opening and test crack direction.

**Figure 25 materials-16-05217-f025:**
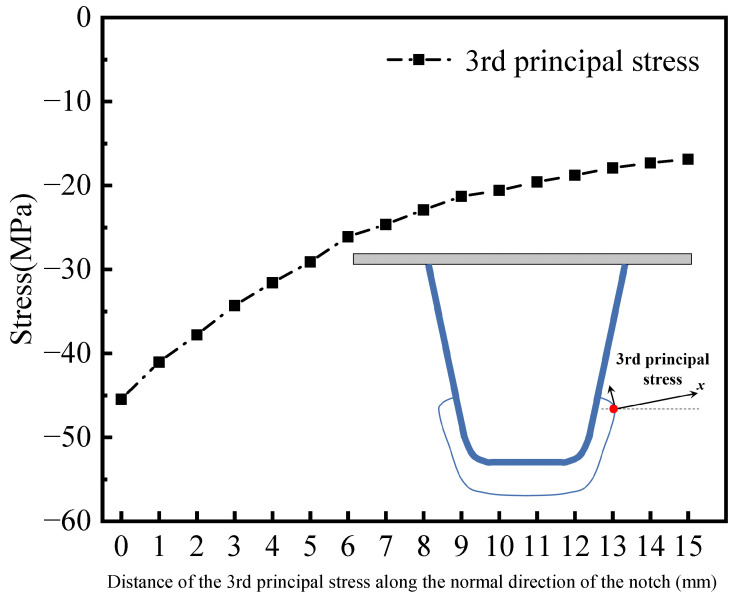
Variation in the third principal stress along the normal X-axis direction of the opening.

**Figure 26 materials-16-05217-f026:**
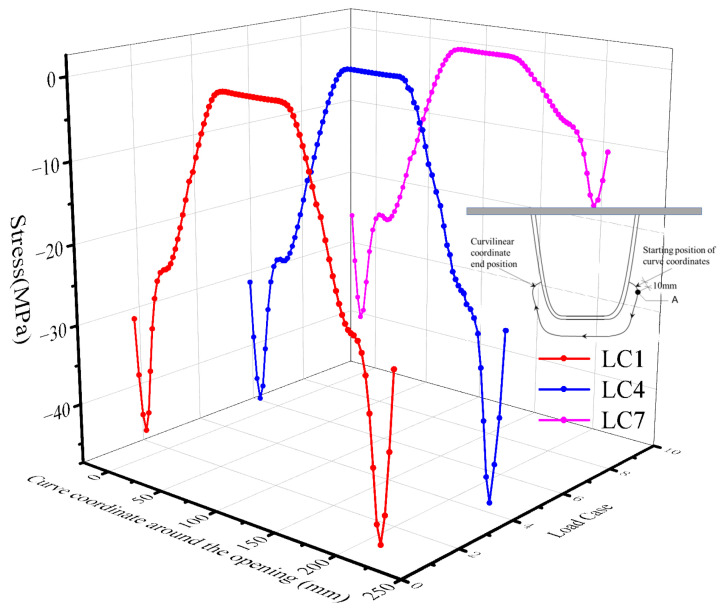
Stress distribution at free edge of opening.

**Figure 27 materials-16-05217-f027:**
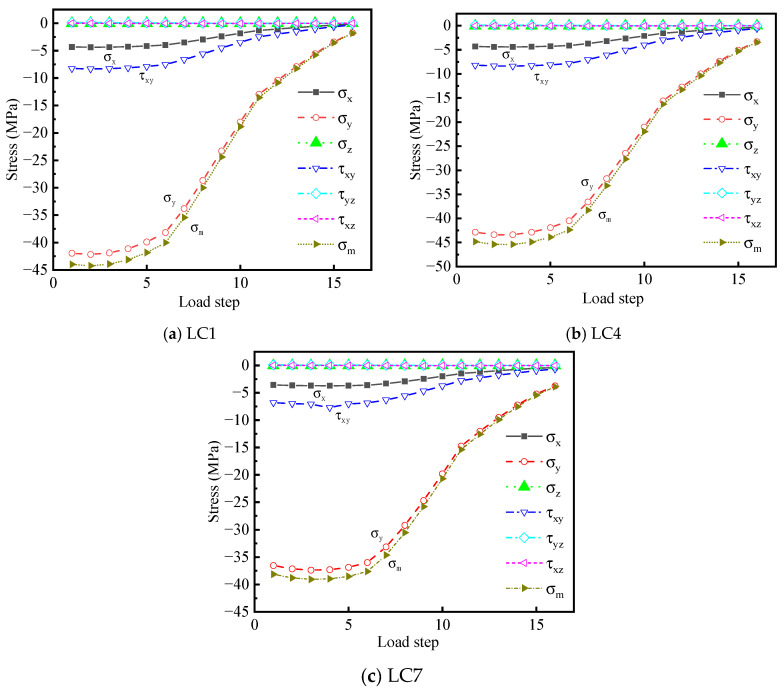
Stress cycle curves of point A under three key load cases.

**Figure 28 materials-16-05217-f028:**
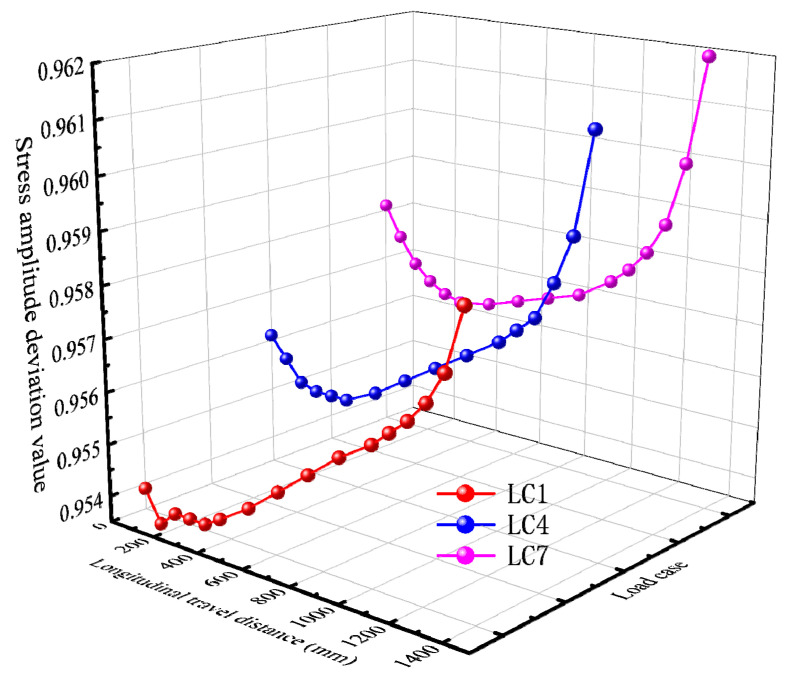
Stress amplitude deviation distribution.

**Figure 29 materials-16-05217-f029:**
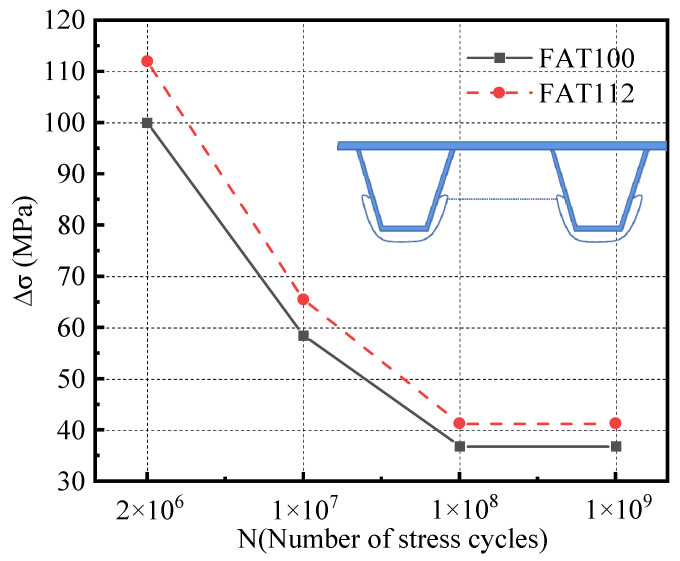
S–N curves for fatigue assessment of diaphragm details.

**Table 1 materials-16-05217-t001:** Comparison between measured stresses around diaphragm’s arc opening and theoretical calculation values.

Measuring Point ID Number	Actual Values (MPa)	Numerical Simulation Values (MPa)	Ratio of Actual Values to Numerical Simulation Values
144	−59.8	−56.9	1.05
145	−28.7	−29.9	0.96
147	−37.3	−38.9	0.96
148	−33.2	−31.6	1.05
149	−24.5	−22.5	1.09
150–152	−28.9	−21.4	1.35
154–156	−19.8	−11.5	1.72
157–158	−15.2	−13.1	1.16
159	−33.0	−35.5	0.93
160	−77.3	−72.2	1.07
161	−33.0	−33.7	0.98
164	−28.6	−31.4	0.91
165	−67.7	−63.3	1.07

**Table 2 materials-16-05217-t002:** Comparison between measured stresses of U-rib and theoretical calculation values.

Measuring Point ID Number	Actual Values (MPa)	Numerical Simulation Values (MPa)	Ratio of Actual Values to Numerical Simulation Values
U02	35.8	33.1	1.08
U05	43.1	39.2	1.10
U09	3.1	3.0	1.03
U11–U13	9.5	10.4	0.91
U15	46.6	40.5	1.15
U16–U18	17.5	18.8	0.93
U19–U21	14.2	12.9	1.10
U22	46.4	44.2	1.05
U23–U25	8.2	8.9	0.92

**Table 3 materials-16-05217-t003:** Moving load at the longitudinal position.

Load Step (*i*)	Longitudinal Coordinate of Moving Load Center (*m*)	Load Step (*i*)	Longitudinal Coordinate of Moving Load Center (*m*)
1	0 (Middle diaphragm midpoint)	11	1
2	0.07	12	1.07
3	0.14	13	1.14
4	0.21	14	1.21
5	0.28	15	1.28
6	0.35	16	1.35 (Midpoint of side diaphragm HGB3)
7	0.48	17	1.6
8	0.61	18	1.85
9	0.74	19	2.1
10	0.87		

**Table 4 materials-16-05217-t004:** Comparison of nominal stress under different meshing types.

No.	Mesh Size (mm)	Total Stress (MPa)	In-Plane Stress (MPa)	Out-of-Plane Stress (MPa)
1	3.0	−46.86	−46.17	−0.69
2	2.0	−49.35	−48.62	−0.73
3	1.0	−49.43	−48.69	−0.74

**Table 5 materials-16-05217-t005:** Values of principal stress under LC4 load case.

Stress	Nominal Stress at 2 mm	Nominal Stress at 6 mm	Nominal Stress at 10 mm	Nominal Stress at 13 mm	Nominal Stress at 15 mm	HSS
Stress value (MPa)	−40.89	−33.27	−28.07	−24.47	−21.93	−44.70

**Table 6 materials-16-05217-t006:** Calculation results of stress amplitude deviation δ.

Load Case	Load Step									
	1	2	3	4	5	6	7	8	9	10
LC1	0.9539	0.9533	0.9536	0.9536	0.9536	0.9538	0.9542	0.9547	0.9552	0.9557
LC4	0.9561	0.9557	0.9553	0.9552	0.9552	0.9552	0.9555	0.9559	0.9563	0.9567
LC7	0.9581	0.9575	0.957	0.9567	0.9565	0.9564	0.9565	0.9567	0.9569	0.9571
Load Case	Load Step									
	11	12	13	14	15	16				
LC1	0.9561	0.9564	0.9567	0.9571	0.9577	0.9589				
LC4	0.9571	0.9574	0.9577	0.9584	0.9593	0.9612				
LC7	0.9575	0.9578	0.9582	0.9588	0.9600	0.9620				

**Table 7 materials-16-05217-t007:** Fatigue life assessment of the opening.

Fatigue Level	Constant Amplitude Fatigue Limit (MPa)	Cut-Off Stress Amplitude (MPa)	Constant m	Constant C	Fatigue Life (Year)
FAT100	58.5	36.9	5	6.84 × 10^15^	7.00
FAT112	65.5	41.3	5	1.21 × 10^16^	12.38

## Data Availability

All data generated or analyzed during this study are included in this article. All data included in this study are available upon request through contact with the corresponding author.

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
