# Peer review of "Study on Fatigue Cracking of Diaphragm’s Arc Opening of OSD in Steel Bridges by Using Biaxial Stress Method"

_materials, 2023, doi:10.3390/ma16155217_

Round 1

Reviewer 1 Report

Manuscript ID: materials-2417639

Title: Study on Fatigue Cracking of Diaphragm’s Arc Opening of 1 OSD in Steel Bridges by Using Biaxial Stress Method

Orthotropic Steel Deck (OSD) is frequently used due to its light weight, high strength properties and excellent stress properties in Long span steel bridges. In this manuscript, the cracking mechanism of arc-opening OSD, which has not been investigated before, is investigated and fatigue cracks are observed during the related fatigue tests. Therefore, this manuscript examines the fatigue performance of the arc opening area of ​​the diaphragm based on the fatigue cracking phenomenon. In this manuscript, stress response of the diaphragm’s arc opening area under the local action of the moving wheel loads was modelled and analyzed using the FEM software known as ABAQUS. Using the HSS method, the hot spot stress has calculated, the stress composition of the key area and arc opening fatigue crack control stresses were analyzed.

In the research, the cracking mechanism of arc-opening OSD was investigated and fatigue cracks were observed during fatigue tests. Therefore, fatigue cracking, which has not been studied before, is investigated in this paper. Therefore, it is an original work.

Acceptable with minor correction.

There are some typos. For example, in some places units were written adjacent to numbers.

Literature spelling rules should be observed.

Long sentences should be shortened to make them more understandable.

Author Response

We would like to thank you for your careful reading, helpful comments, and constructive suggestions, which has significantly improved the presentation of our manuscript.

There are some typos. For example, in some places units were written adjacent to numbers. Literature spelling rules should be observed. Long sentences should be shortened to make them more understandable.

Reply: Thank you for your comments. We have re-written some sentences according to your suggestion.

Reviewer 2 Report

Manuscript to be corrected:

1. Authors need to explain the symbols after each formula

2. The abstract should clearly highlight the new point of the article

3. The authors need to specify how the model of the applied load to get the results as shown in Figure 21?

4. Some drawings are too small, need to be shown more clearly: 4b, 11, 17,...

5. The manuscript needs to clarify the physical meaning of some calculation results

6. The manuscript should cite several articles on cracks:

- "Thermal buckling analysis of cracked functionally graded plates"

- "Finite element modeling of free vibration of cracked nanoplates with flexoelectric effects"

- "Phase field model for fracture based on modified couple stress"

Minor editing of English language required

Author Response

We would like to thank you for your careful reading, helpful comments, and constructive suggestions, which has significantly improved the presentation of our manuscript.

  1. Authors need to explain the symbols after each formula

Reply: We have added the definitions of the symbols after each formula in the manuscript.

  1. The abstract should clearly highlight the new point of the article

Reply: We have revised the abstract. The revised abstract reads:

The changes in loading position have a significant impact on the stress field of each vulnerable area of the orthotropic steel deck(OSD). The arc opening area of the diaphragm and the connecting area between the U-rib and the diaphragm under the moving load is prone to fatigue cracking. By comparing the stress responses under different methods, the Hot Spot Stress(HSS) method is used as the main stress extraction method in fatigue performance evaluation. The control stress of fatigue cracking was analyzed by comparing the direction of the principal stress field with the crack direction in the experiment. According to the stress amplitude deviation under the biaxial stress state, a set of methods for evaluating the effects of in-plane biaxial fatigue is developed. The improved luffing fatigue assessment S-N curve is applied to analyze the fatigue life of the diaphragm's arc opening area. The results shows that when the moving load is exactly above the connection of the deck and the web of U-rib on one side, it is the most unfavorable position in the transverse direction, and the diaphragm is mainly under the in-plane stress state. The longitudinal range of the stress influence line of the arc opening is approximately twice the diaphragm spacing. 2~3 stress cycles are caused by one fatigue load. The fatigue crack control stress is the principal stress tangent to the arc opening’s edge in this area. The normal direction of the principal stress in the model test is roughly consistent with the crack initiation direction. The variation in the stress amplitude deviation in this area is caused by changes in the action position of the moving load. When the moving load is at a certain distance from the concerned diaphragm, is reduced to zero, implying that the in-plane fatigue effect is the greatest in this area.

  1. The authors need to specify how the model of the applied load to get the results as shown in Figure 21?

Reply: We have adjusted the order of the figures in the article. Figure 21 is now Figure 22. The three load cases shown in Figure 22 are all described in Section 4.1, and we have indicated the source of the load cases before the analysis in Figure 22.

  1. Some drawings are too small, need to be shown more clearly: 4b, 11, 17,...

Reply: We have replaced these figures with clear ones.

  1. The manuscript needs to clarify the physical meaning of some calculation results

Reply: Thank you for your careful check. We have explained the meaning of the calculation results in the manuscript.

  1. The manuscript should cite several articles on cracks:

- "Thermal buckling analysis of cracked functionally graded plates"

- "Finite element modeling of free vibration of cracked nanoplates with flexoelectric effects"

- "Phase field model for fracture based on modified couple stress"

Reply: We have read and cited several articles related to cracks. We will learn and draw on more in our future research writing.

Reviewer 3 Report

This research work characterized the stress distributions on OSD structure experimentally and compared them with the finite element methods. Another objective is to analyze and predict the fatigue cracking behavior in these bridge structures.

However, the manuscript has significant issues that need to be addressed. It is excessively long as a research paper, difficult to understand, and follows – 32 pages and 28 figures. The contents need to be organized in a clear manner. The reviewer was having a hard time understanding the content and the discussions. The OSD components need to be drawn and described in detail at the beginning of the manuscript. Variables need to be defined when first discussed.

In the stress analysis, the reviewer is not clear about which stress is being discussed.  The reviewer cannot evaluate the contents of this manuscript in its current form.

Here are some questions (but not all) that the author should be addressed.

·        HSS was not defined until line 86. Should define it when first mentioned.

·        Please use engineering notation throughout the manuscript.

·        Table 1. “Number of measuring points” – Should it be “measuring point ID number”?

·        “Measured values”. What kind of stress values are they? Principal stresses? Is this the stress calculated from Hooke’s law from the strain gauge data? Do “Theoretical calculation values” mean simulated values? Please be specific.

·        Define “FEM”.

·        Table 2 – where are the “Uxx” measuring points?

·        Line 305- Where is “6#” first defined in the manuscript? Need schematic drawings to define it.

·        Please highlight cracks in Figure 7, the reviewer cannot see them.

·        FEM model mesh size, node type, step size, and other simulation conditions need to be defined in the Method section.

·        What do the numbers in Figure 14 (305 and 900) mean?

·        The reviewer does not understand Figure 15.

·        What kind of stress is shown in Figure 16? Min principal stress? Should max principal stress (tensile) that drives crack growth?

·        Figure 18, the mesh in the bottom of the U seems to have elements significantly larger than the rest of the model. Will that be a concern?

·        What is “nominal stress” mentioned in the manuscript?

·        Are the welding materials different from the diaphragm? Is the model using the same elastic modulus for the weld?

·        Figure 26 – where is point A? Need drawings to show this.

·        Define back, front, in-plane, and out-of-plane stress when first mentioned. Schematic drawing will be useful

·        Figure 27. What is the load circle axis?

·        Figure 23. Both the 2nd and 3rd principal stress labels are pointing at the same purple line. The reviewer cannot identify the crack in the photo.

This manuscript needs to reorganize in a clear manner.

Author Response

We would like to thank you for your careful reading, helpful comments, and constructive suggestions, which has significantly improved the presentation of our manuscript.

  1. Table 1. “Number of measuring points” – Should it be “measuring point ID number”?

Reply: Thank you for your consideration. We have changed the “Number of measuring points” into “Measuring Point ID Number” for better expression.

  1. “Measured values”. What kind of stress values are they? Principal stresses? Is this the stress calculated from Hooke’s law from the strain gauge data? Do “Theoretical calculation values” mean simulated values? Please be specific.

Reply: Thank you for your suggestion. The “Measured values” have been changed into the “Actual values”. It represent the maximum absolute principal stress. And it is calculated from the strain gauge data by the Hooke’s law. “Theoretical calculation values” means “Numerical Simulation values”. We have changed “Theoretical calculation values” into “Numerical Simulation values” for better expression of meaning.

  1. Define “FEM”.

Reply: FEM stands for the Finite Element Method. We have defined it when it was first mentioned in the manuscript.

  1. Table 2 – where are the “Uxx” measuring points?

Reply: We have added another layout of the stress measuring points in Fig.6. We are sorry for our negligence.

  1. Line 305- Where is “6#” first defined in the manuscript? Need schematic drawings to define it.

Reply: “6#” means the 6#Rib as shown in Fig.6. We have made corrections in the manuscript.

  1. Please highlight cracks in Figure 7, the reviewer cannot see them.

Reply: We have adjusted the order of the figures in the article. Originally, Figure 7 is now Figure 8. We have highlighted the cracks in Fig.8.

  1. FEM model mesh size, node type, step size, and other simulation conditions need to be defined in the Method section.

Reply: Thank you for the above suggestion. We have added details of the finite element model in the Method section.

  1. What do the numbers in Figure 14 (305 and 900) mean?

Reply: We have adjusted the order of the figures in the article. Figure 14 is now Figure 15. 305 mm is the transverse length of the contact area after the wheel load diffused to the bridge deck at 45 degree. After scaling the lateral wheelbase 1800mm in a ratio of 1:2, the lateral wheelbase is taken as 900mm. Section 4.2 provides a detailed introduction to the selection and application of fatigue loads.

  1. The reviewer does not understand Figure 15.

Reply: Figure 15 is now Figure 16. Figure 16 is a longitudinal schematic of the fatigue load cases, corresponding to the transverse schematic shown in Figure 15. Starting from the bridge deck directly above the middle transverse partition (HGB2), move towards the side transverse partition (HGB3). The numbers 1 to 19 above represent the corresponding load step numbers for the longitudinal coordinates. The specific longitudinal coordinates of the load center for each load step are detailed in Table 15.

  1. What kind of stress is shown in Figure 16? Min principal stress? Should max principal stress (tensile) that drives crack growth?

Reply: Figure 16 is now Figure 17. Figure 17 shows the distribution of the third principal stress, which is the minimum principal stress. The positive value of the principal stress is tensile, the negative value of the principal stress is compressive. It can be seen that the third principal stress at the stress concentration point in Fig.17 is compressive stress. According to the simulation results, within a certain range of fatigue load, the second and third principal stresses near the stress concentration point are both negative, indicating that the point is in a state of biaxial compression within a certain range. Due to the large absolute values of the third principal stress in various nearby areas, fatigue performance evaluation is conducted using the third principal stress as a representative. So we chose the third principal stress to display the stress concentration point, as shown in Figure 17.

  1. Figure 18, the mesh in the bottom of the U seems to have elements significantly larger than the rest of the model. Will that be a concern?

Reply: Figure 18 is now Figure 19. Thank you for your careful check. The larger mesh in the bottom of the U-Rib is a software display issue. It does not affect the numerical simulation results. We have replaced these figures to avoid misunderstandings.

  1. What is “nominal stress” mentioned in the manuscript?

Reply: We use the nominal stress method to perform grid independence checks in finite element analysis to determine the appropriate grid partitioning method. And in Section 4.3, the stress responses of two different stress extraction methods, the hot spot stress method and the nominal stress method, were compared. The analysis shows that the hot spot stress method is more suitable for control stress analysis and fatigue performance evaluation. Due to the length of the article, we did not include a section on the nominal stress method in the article.

  1. Are the welding materials different from the diaphragm? Is the model using the same elastic modulus for the weld?

Response: In the fatigue model experiment, the welding material and the diaphragm material are the same, and their elastic modulus is also the same. Therefore, when conducting numerical simulations, we adopted a common node approach to establish the model at the weld joints between the U-rib and the diaphragm.

  1. Figure 26 – where is point A? Need drawings to show this.

Reply: Figure 26 is now Figure 27. Point A is shown in Fig.18. The detailed introduction of point A is on line 407.

  1. Define back, front, in-plane, and out-of-plane stress when first mentioned. Schematic drawing will be useful

Reply: We have defined the back, front, in-plane, and out-of-plane stress when first mentioned on line 216 and line 548.

  1. Figure 27. What is the load circle axis?

Reply: Figure 27 is now Figure 28. We did not find the load circle axis in Figure 28 during the check.

  1. Figure 23. Both the 2nd and 3rd principal stress labels are pointing at the same purple line. The reviewer cannot identify the crack in the photo.

Reply: Figure 23 is now Figure 24. The purple line represent the 2nd principal stress, the yellow line in the center represents the 3rd stress. We have highlighted the yellow line in Fig.24(a) and the crack in Fig.24(b) for better expression.

  1. This manuscript needs to reorganize in a clear manner.

Reply: We have reorganized our manuscript.

Round 2

Reviewer 2 Report

This work is good now